# CRPO: Character-centric Group Relative Policy Optimization
# for Role-aware Reasoning in Role-playing Agents

**Yihong Tang** [1][2]  **Kehai Chen** [1]  **Liang Yue** [1]  **Benyou Wang** [3][2]  **Min Zhang** [1][2]

## Abstract

Recent advancements in Reinforcement Learning (RL), particularly Group Relative Policy Optimization (GRPO), have significantly enhanced the reasoning capabilities of Large Language Models. However, applying these problem-centric optimization methods to role-playing agents often leads to a loss of character fidelity and style collapse, as they prioritize context-specific utility over persona alignment. To address this, we propose Character-Centric Group Relative Policy Optimization (CRPO), a framework designed to realign RL objectives with the role-playing task. CRPO improves character distinctiveness through three mechanisms: decoupling task logic from stylistic rewards to resolve gradient conflicts, dynamically adapting optimization constraints based on character complexity, and utilizing generic responses as negative baselines to prevent the model from reverting to a common distribution. Extensive experiments demonstrate that CRPO outperforms existing methods in consistency, emotion and others. Our code is available at https://github.com/Toyhom/CRPO.

## 1. Introduction

Recently, efficient Reinforcement Learning (RL) algorithms, represented by Group Relative Policy Optimization (GRPO) (Shao et al., 2024), are rapidly becoming the core drivers shaping the reasoning capabilities of Large Language Models (LLMs), catalyzing a series of improvements such as DAPO (Yu et al., 2025b) and GSPO (Zheng et al.,

[1]Institute of Computing and Intelligence, Harbin Institute of Technology, Shenzhen, China [2]Shenzhen Loop Area Institute (SLAI), Shenzhen, China [3]The Chinese University of Hong Kong, Shenzhen, China. Correspondence to: Kehai Chen <chenkehai@hit.edu.cn>.

*Proceedings of the 43rd International Conference on Machine Learning*, Seoul, South Korea. PMLR 306, 2026. Copyright 2026 by the author(s).

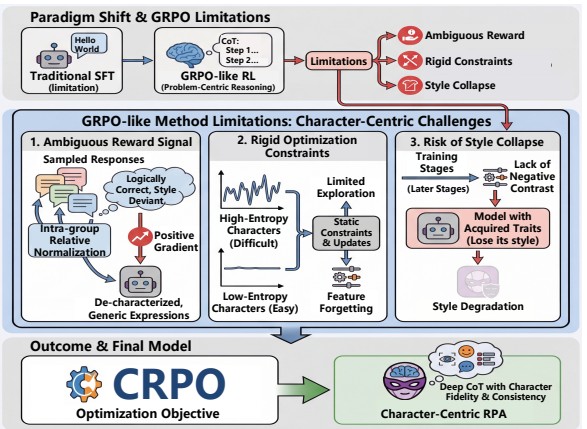

*Figure 1.* Problem-centric optimization suffers from ambiguous rewards, rigid constraints, and style collapse, which impede consistent persona alignment.

2025a). This paradigm shift also propels the field of Role-Playing Agents (RPAs) (Wang et al., 2024a) through a transformation from *behaviorist imitation* to *cognitivist reasoning*. Traditional Supervised Fine-Tuning (SFT), representative of the former, often focuses on imitating the shallow linguistic style of characters. Conversely, recent preference-optimization methods have explored global persona faithfulness, e.g., optimizing active and passive character constraints through DPO-style objectives (Peng & Shang, 2024). In parallel, emerging GRPO-based methods also introduce verifiable rewards—such as cognitive focus (Tang et al., 2026), keywords (Wang et al., 2025c), and emotion (Wang et al., 2025f)—to incentivize models to internalize character cognitive patterns through deep Chains of Thought (CoT), preliminarily enabling the *Role-aware Reasoning* (Feng et al., 2025; Tang et al., 2025b).

However, existing GRPO-like methods exhibit a significant limitation: these techniques primarily reward LLMs for generating isolated, context-oriented responses (i.e., *problem-centric*). This reduces the model's incentive to maintain character fidelity, indicating a lack of a *character-centric* optimization objective. As illustrated in Figure 1, this limitation manifests in three main challenges: (1) **Ambiguous Reward Signal**: Standard GRPO relies on intra-group relative normalization. When a sampled group collectively deviates from the character style but remains

logically correct, the relative advantage still assigns positive gradients, inducing the model to learn de-characterized, generic expressions. (2) **Rigid Optimization Constraints**: The difficulty of fitting different characters (entropy) varies significantly. Character-agnostic constraints and update mechanisms fail to adapt to this dynamic variation, limiting exploration for high-entropy characters and causing feature forgetting in low-entropy characters. (3) **Risk of Style Collapse**: Due to the lack of explicit negative sample contrasts, models often fall into style degradation during the later stages of training, inadvertently weakening the retention of acquired high-accuracy traits. Addressing these challenges is crucial for improving character consistency and the depth of role-aware reasoning, especially when facing complex instructions that require balancing task logic with persona style.

To this end, we propose **Character-Centric Group Relative Policy Optimization (CRPO)**, aiming to reconstruct the optimization mechanism of GRPO in role-playing. Our framework consists of three key components: (1) **Dual-Stream Advantage Estimation** to address ambiguous reward signals by decomposing rewards into task and style streams, utilizing distinct normalization strategies for each; (2) **Entropy-Aware Adaptive Exploitation** to resolve rigid optimization constraints by dynamically regulating update intensities based on character uncertainty; and (3) **Contrastive Anchor Sampling** to mitigate style collapse by introducing generic responses as negative anchors to enforce character-specific learning.

The main contributions of this paper are summarized as follows:

(1) To the best of our knowledge, we are the first to propose a character-centric GRPO-like framework, CRPO, for role-playing, recalibrating the construction of RPAs from the optimization objectives.

(2) Under the CRPO framework, we propose the synergistic operation of Dual-Stream Advantage Estimation, Entropy-Aware Adaptive Exploitation, and Contrastive Anchor Sampling. These methods alleviate the challenges of reward ambiguity, training instability, and style collapse.

(3) Extensive experiments illustrate that CRPO significantly enhances the character consistency, hallucination mitigation and robustness of role-playing agents.

## 2. Related Work

### 2.1. Role-playing Agents

The pursuit of emotionally intelligent and consistent virtual interactions is driving Role-Playing Agents (RPAs) beyond mere *behavioral imitation* toward *cognitive simulation*. The former focused on capturing surface-level linguistic patterns

through prompt engineering (Tang et al., 2025a; Duan et al., 2025), SFT (Wang et al., 2024a; Zhou et al., 2024a; Yu et al., 2024; Lu et al., 2024; Li et al., 2025; Yang et al., 2025c), retrieval-augmented generation (Chen et al., 2025) and formalized character logic (Peng & Shang, 2025; Peng et al., 2026). In parallel, the Sotopia series (Zhou et al., 2024b; Wang et al., 2024b; Yu et al., 2025a) explores the training of social agents in interactive environments, which is complementary to role-playing. However, relying solely on surface patterns often leads to logical inconsistencies and character drift. This limitation has spurred a shift toward explicit cognitive simulation.

Recent studies, including TBS (Zhang et al., 2024), CoSER (Wang et al., 2025d), REDEN-R1 (Wang et al., 2025f), CPO (Ye et al., 2025), MOA (Liao et al., 2025) and Character-R1 (Tang et al., 2026), now pioneer the modeling of internal thought processes. By leveraging techniques such as CoT (Wei et al., 2022), distillation and RL, these works preliminarily achieve role-aware reasoning. Nevertheless, most of these attempts directly apply general-purpose RL algorithms without tailoring the optimization dynamics to the nuances of persona maintenance.

### 2.2. Group Relative Policy Optimization and Variants

Group Relative Policy Optimization (GRPO) (Shao et al., 2024) removes value networks for efficiency (Jiang et al., 2025) but faces limitations prompting refinements: (1) **Credit Assignment**. To refine coarse rewards, OAR (Li et al., 2026), $\lambda$-GRPO (Wang et al., 2025e), GSPO (Zheng et al., 2025a) and GTPO (Simoni et al., 2025) optimize token-level weights or suppress misleading updates via counterfactual perturbations, entropy control, and likelihood constraints. (2) **Signal Enhancement**. Addressing sparse or noisy signals, NGRPO (Nan et al., 2025), SGPO (Lee et al., 2025), AMIR-GRPO (Yari & Koto, 2026), RiskPO (Ren et al., 2025), KRPO (Wang et al., 2025b), GDPO (Liu et al., 2026), EDGE-GRPO (Zhang et al., 2025a), and AEPO (Wang et al., 2025a) leverage error transformation, risk-based ranking, Kalman dynamics, or entropy-driven exploration to calibrate advantage estimation. (3) **Stability & Efficiency**. To reduce gradient conflicts, DaGRPO (Xie et al., 2025), SFPO (Wang et al., 2025g), and Dr. GRPO (Liu et al., 2025) employ distinctiveness sampling, dual-layer updates, or length normalization strategies. (4) **Guided Exploration**. Scaf-GRPO (Zhang et al., 2025b), PTA-GRPO (Dou et al., 2025), and Training-Free GRPO (Cai et al., 2025) incorporate hierarchical prompting or context-based guidance to navigate complex tasks. Despite these variants achieving strong performance in mathematical and code reasoning tasks, they largely adhere to a problem-centric optimization paradigm that maximizes objective utility and logical correctness for input instructions. This singular perspective falls short for role-playing tasks, as

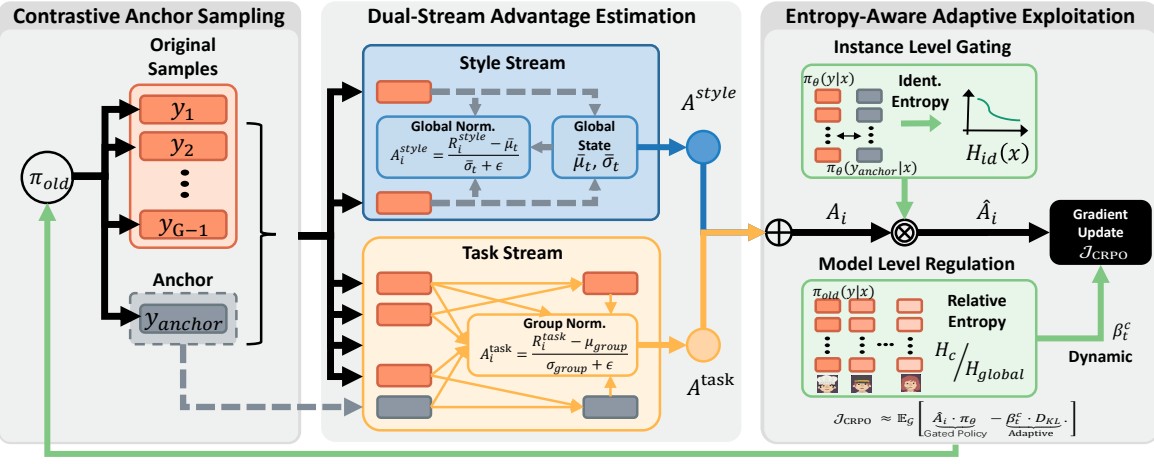

*Figure 2.* The overall framework of CRPO. The method synergizes three core mechanisms to enhance role-playing: (1) Contrastive Anchor Sampling prevents style collapse by introducing generic negative samples; (2) Dual-Stream Advantage Estimation decouples task and style rewards to resolve optimization conflicts; and (3) Entropy-Aware Adaptive Exploitation dynamically regulates gradient updates based on character-specific uncertainty and difficulty.

it overlooks the core need to be character-centric—both solving tasks and enacting a persona with specific cognitive patterns and linguistic styles. Based on this shift in perspective, CRPO addresses three key deficiencies in the GRPO paradigm.

## 3. Method

The CRPO framework aims to redirect the optimization objective of reinforcement learning from generic "problem-centric" to "character-centric" through three synergistic components. Addressing the issue of ambiguous reward signals, we introduce **Dual-Stream Advantage Estimation**, decoupling task-related relative rewards from character-related absolute rewards.

Addressing the heterogeneity in fitting difficulty across different characters, we design **Entropy-Aware Adaptive Exploitation** to dynamically adjust the optimization constraints and update intensity for each character. Addressing the risk of style degradation in the later stages of training, we propose a **Contrastive Anchor Sampling** strategy, preventing the policy from collapsing into a generic distribution by forcibly injecting negative samples into the sampling group. Figure 2 illustrates the overall architecture of CRPO.

### 3.1. Preliminary

GRPO is a PPO variant that removes the Value Function. It estimates the advantage function by normalizing intra-group rewards for multiple sampling results generated from a single prompt, thereby reducing resource consumption and stabilizing training. Formally, given a prompt $x$ and a policy model $\pi_\theta$, GRPO samples a set $\mathcal{G} = \{y_i\}_{i=1}^G$, containing $G$ outputs from the old policy $\pi_{old}$. For each output $y_i$,

a corresponding scalar reward $R_i$ is calculated, and the advantage function $A_i$ is computed based on intra-group statistics:

$$A_i = \frac{R_i - \text{mean}(\{R_1, \ldots, R_G\})}{\text{std}(\{R_1, \ldots, R_G\}) + \epsilon}, \tag{1}$$

where $\epsilon$ is a smoothing term to prevent division by zero. The objective function $\mathcal{J}_{GRPO}$ combines a clipping mechanism for the importance sampling ratio $\rho_i = \pi_\theta(y_i|x)/\pi_{old}(y_i|x)$ with a KL divergence penalty:

$$\mathcal{J}_{GRPO}(\theta) = \mathbb{E}_{x \sim \mathcal{D}, \{y_i\} \sim \pi_{old}} \left[ \frac{1}{G} \sum_{i=1}^G \Big( \min(\rho_i A_i, \right.$$
$$\left. \text{clip}(\rho_i, 1-\varepsilon, 1+\varepsilon)A_i) - \beta \mathbb{D}_{KL}(\pi_\theta || \pi_{ref}) \Big) \right], \tag{2}$$

where $\mathcal{D}$ denotes the training dataset used for optimization, $\pi_{\text{ref}}$ is the fixed reference model, $\varepsilon$ is the clipping threshold for importance ratios, and $\beta$ is the KL penalty coefficient.

In role-playing reasoning tasks, we require the model-generated response $y$ to contain a clear Chain of Thought $y_{CoT}$ and a character response $y_{ans}$, denoted as $y = y_{CoT} \oplus y_{ans}$. Since role-playing is typically an open-ended generation task lacking a single objective standard answer, this not only increases the difficulty of reward design but also facilitates reward hacking. To mitigate this, our reward system is built upon Character-R1, introducing two types of verifiable rewards: (1) **Cognitive Focus Reward**: Requiring the model to explicitly state and elaborate on the current cognitive focus within $y_{CoT}$ as a checkable objective fact. (2) **Reference-Guided Reward**: Utilizing overlap-based metrics to guide the model closer to the target reference style while mitigating overfitting risks.

## 3.2. Dual-Stream Advantage Estimation

In role-playing, logical correctness is relative (dependent on context difficulty), whereas character style is absolute (adhering to a fixed profile). Direct mixed normalization of the two leads to ambiguous gradient directions. Therefore, we propose **Dual-Stream Advantage Estimation**, decomposing the total reward into task reward $R^{task}$ and style reward $R^{style}$ for independent processing.

**Task Stream.** To adapt to varying input problem difficulty, we adopt intra-group relative normalization. For the $i$-th sample in the group, its task advantage $A_i^{task} = \frac{R_i^{task} - \mu_{\mathcal{G}}^{task}}{\sigma_{\mathcal{G}}^{task} + \epsilon}$. where $\mu_{\mathcal{G}}^{task}$ and $\sigma_{\mathcal{G}}^{task}$ are the mean and standard deviation of task rewards within the sampling group $\mathcal{G}$, respectively. This relative evaluation ensures that logically relatively better responses receive positive incentives even when the generation quality of the entire group is limited by high-difficulty problems.

**Style Stream.** To stabilize style rewards across batches, we introduce global normalization based on historical statistics. Style rewards reflect the degree of fit between the model and a specific character $c$ and should not fluctuate with batch sample quality. We maintain the exponential moving average of the historical reward mean $\bar{\mu}_{c,t}$ and variance $\bar{\sigma}_{c,t}$ for that character as normalization parameters. The style advantage $A_i^{style} = \frac{R_i^{style} - \bar{\mu}_{c,t}}{\bar{\sigma}_{c,t} + \epsilon}$. The final advantage function is a weighted synthesis of both: $A_i = \lambda A_i^{task} + (1 - \lambda) A_i^{style}$, where $\lambda$ is a balancing coefficient. This mechanism effectively decouples the optimization paths for logical capability and style performance.

## 3.3. Character Entropy-Aware Adaptive Exploitation

Since the knowledge of Large Language Models primarily originates from pre-training, forcibly requiring the model to undergo character style transfer on semantically ambiguous samples may destabilize the pre-training distribution. We propose an **Entropy-Aware Adaptive Exploitation** mechanism to dynamically regulate optimization constraints and update strategies at both instance-level and model-level.

**Instance Level.** We define Character Identification Entropy to quantify the model's confidence in distinguishing role-playing from generic responses. For input $x$, we calculate the character generation probability ratio $p_r(x)$ and the corresponding binary entropy $H_{id}(x)$:

$$H_{id}(x, y) = - \left[ p_r \log p_r + (1 - p_r) \log(1 - p_r) \right], \quad (3)$$

where $p_r(x, y) = \frac{\pi_\theta(y|x)}{\pi_\theta(y|x) + \pi_\theta(y_{anchor}|x)}$, $y$ is the character response generated by $\pi_{old}$, and $y_{anchor}$ is the response generated by the model after removing character instructions (retaining only the dialogue context). In this calculation, we treat sequence probability as overall confidence. A higher $H_{id}(x)$ indicates that the model struggles to define the style attribution of the current response. Accordingly, we re-weight the advantage function $\hat{A}_i = A_i \cdot (1 - \gamma H_{id}(x))$, where $\gamma > 0$ is a control coefficient. This mechanism acts as a safety valve; when the model is in a state of high uncertainty, the gradient signal is significantly suppressed, thereby achieving cautious optimization.

**Model Level.** To adapt to the intrinsic fitting difficulty of different characters, we calculate the average information entropy $H_c$ of the reference model $\pi_{ref}$ on character $c$'s data and define its ratio $r_H = H_c / H_{global}$ relative to the global average entropy $H_{global}$. For high-entropy (complex, variable, and hard-to-learn) characters, we relax the KL divergence target value $d_{targ}^c$ via an adaptive formula: $d_{targ}^c = d_{targ}^{base} \cdot \text{clamp}(r_H, \delta_{min}^{KL}, \delta_{max}^{KL})$. Subsequently, a proportional-integral controller (Ziegler et al., 2019) is used to dynamically adjust the penalty coefficient $\beta_t^c$ based on the observed KL deviation over time:

$$e_t^c = \text{clip}\left( \frac{D_{\text{KL},t}}{d_{\text{targ},t}^c} - 1, -\delta_{bound}, \delta_{bound} \right), \quad (4)$$

$$\beta_{t+1}^c = \beta_t^c \cdot (1 + K_p \cdot e_t^c). \quad (5)$$

where $e_t^c$ is the truncated error term, $\delta_{bound}$ is the error truncation threshold, and $K_p$ is the proportional gain coefficient. In summary, the instance-level mechanism tends to be conservative microscopically (suppressing high entropy), while the model-level mechanism tends to be aggressive macroscopically (relaxing high entropy), synergistically achieving stable and efficient style alignment. For a formal theoretical justification and detailed proofs regarding the Entropy-Aware Adaptive Exploitation mechanism, please refer to Appendix A.1.

## 3.4. Contrastive Anchor Sampling

In GRPO sampling, advantage estimation relies solely on samples generated by the current policy under the same prompt (Wang et al., 2026). This dependency often traps the model in local optima, leading to late-stage degeneration where the generated content drifts toward a generic, non-character style. Therefore, we propose a **Contrastive Anchor Sampling** strategy. By forcibly injecting negative samples into the sampling group, we utilize implicit contrastive learning to prevent the policy from collapsing into a generic distribution. Specifically, we construct a mixed sampling group $\mathcal{G} = \{y_1, \ldots, y_{G-1}, y_{anchor}\}$, where the first $G - 1$ samples are generated normally by policy $\pi_{old}$, while $y_{anchor}$ is generated by system prompts with character settings removed, representing a "non-character" distribution. The introduction of negative anchors

*Table 1.* Details of the specialized role-playing models.

| Model | Backbone | Data Scale | Method |
|---|---|---|---|
| CharacterGLM-6B (Zhou et al., 2024a) | ChatGLM2-6B | 1k | SFT |
| Peach-9B-8k-Roleplay (ClosedCharacter, 2024b) | Yi-1.5-9B | 100k | SFT |
| Llama-3.1-8B-RoleMRC (Lu et al., 2025) | Llama-3.1-8B | 107.7k | SFT, DPO |
| Haruhi-Zero-7B (Silk Road, 2024) | Qwen-7B | 120k | SFT |
| Crab (He et al., 2025) | Llama-3.1-8B | 41k | SFT |
| CoSER-Llama-3.1-8B (Wang et al., 2025d) | Llama-3.1-8B | 29.7k | SFT |
| Peach-2.0-9B-8k-Roleplay (ClosedCharacter, 2024a) | Yi-1.5-9B | 100k | SFT, DPO |
| Qwen2.5-7B-RoleMRC (Lu et al., 2025) | Qwen2.5-7B | 107.7k | SFT, DPO |
| Humanish-Roleplay (Gallego, 2024) | Llama-3.1-8B | 269.9k | SFT, DPO |
| **CRPO (Ours)** | – | 650 | RL |

lowers the baseline for rewards, compelling the model to actively maximize the difference between character features and generic features. Since $y_{anchor}$ lacks specific character linguistic traits, its style reward $R_{anchor}^{style}$ is significantly lower than that of character-centric responses. By enforcing intra-group normalization on the style stream (after the global normalization in Section 3.2), the presence of this low score amplifies the advantage values $A^{style}$ of samples with distinct character styles, thereby preventing the policy from collapsing towards a generic distribution during the optimization process.

# 4. Experiments

## 4.1. Experimental Setup

**Benchmarks and Datasets.** To rigorously assess the performance of our model, we employ two established public benchmarks. *(1) CharacterBench* (Zhou et al., 2025), a comprehensive generative evaluation suite, contains 22,859 samples spanning 3,956 distinct characters. It scrutinizes 11 specific dimensions, including memory consistency, personality adherence, and knowledge boundaries. *(2) SocialBench* (Chen et al., 2024) targets social intelligence through a hybrid of multiple-choice and open-ended queries. This dataset includes 500 character profiles and over 30,800 multi-turn exchanges, evaluating aspects such as self-awareness and social preference. Experimentally, we utilize a subset of 650 random samples from CharacterBench for the training phase, while SocialBench is reserved exclusively for testing generalization capabilities.

Refer to Appendix B for extensive details.

**Baselines.** To rigorously evaluate the efficacy of our proposed method, we benchmark it against four distinct categories of models: *(1) Role-playing methods*, which employ specific fine-tuning strategies. This category includes SFT, Neeko (Yu et al., 2024), and RAR (Tang et al., 2025b). Notably, Character-R1 (Tang et al., 2026) is highlighted as a pioneer in role-playing task for introducing verifiable rewards, encompassing cognitive focus and reference guidance. *(2) Specialized role-playing models* trained on extensive character-centric corpora,

specifically Character-GLM, Haruhi-Zero, CoSER, and RoleRMC (refer to Table 1 for specifications). *(3) General RL methods*, comprising a suite of standard algorithms: PPO, REINFORCE++, RLOO, REMAX, GRPO, Dr.GRPO, DAPO, GSPO, GDPO, and OAR. Detailed configurations are provided in Table 2. *(4) Large foundation models*, serving as strong generalist baselines, including Llama-3-70B-Instruct (Meta, 2024), GLM-4 (GLM et al., 2024), Baichuan-NPC (Yang et al., 2025b), GPT-3.5-turbo, GPT-4o (OpenAI, 2024), Qwen2.5-72B-Instruct (Qwen et al., 2025), Gemini-3-pro (Deepmind, 2025), and Claude-4-opus (Anthropic, 2025).

*Table 2.* Summaries of common RL methods for RLHF.

| Method | Key Contribution |
|---|---|
| PPO (Schulman et al., 2017) | Stable policy gradient using clipping. |
| R++ (Hu et al., 2025) | REINFORCE with PPO-style stability, no critic. |
| RLOO (Kool et al., 2019) | Batch-peer comparison for variance reduction. |
| ReMax (Li et al., 2024) | Greedy baseline RL without value networks. |
| GRPO (Shao et al., 2024) | Within-group advantage estimation to cut costs. |
| Dr.GRPO (Liu et al., 2025) | Unnormalized GRPO to reduce bias. |
| DAPO (Yu et al., 2025b) | Dynamic sampling and asymmetric clipping. |
| GSPO (Zheng et al., 2025a) | Sequence-level optimization for stable alignment. |
| GDPO (Liu et al., 2026) | Generative flows for diversity and alignment. |
| OAR (Li et al., 2026) | Outcome-aware fine-grained credit assignment. |
| **CRPO (Ours)** | Character-centric optimization. |

**Implementation.** We implement CRPO using the EasyR1 framework (Zheng et al., 2025b) with Llama-3.2-3B-Instruct (Meta, 2024) and Qwen3-8B (Yang et al., 2025a) as backbones. Further implementation details can be found in Appendix B.3.

## 4.2. Main Results

**Performance on Comprehensive Role-Play Dialogue Generation.** As shown in Table 3, CRPO consistently outperforms all baselines across both Llama-3.2-3B and Qwen3-8B backbones, establishing a new state-of-the-art on CharacterBench. Broadly speaking, compared to SFT baselines such as Neeko and RAR, RL methods represented by CRPO demonstrate remarkable data efficiency, eliciting superior role-playing capabilities with only a fraction of the training data. Notably, CRPO and certain advanced RL baselines even surpass proprietary large foundation models in specific metrics. This validates the efficiency of our character-centric optimization objective and highlights the immense potential of RL in vertical domains.

First, regarding *Persona Consistency*, CRPO overcomes the *alignment tax* observed in general RL methods, where models sacrifice stylistic diversity for logical correctness, often degenerating into generic assistants. By employing Dual-Stream Advantage, CRPO decouples style from logic, maintaining high fidelity without being penalized for the inherent entropy of role-playing. Second, regarding *Knowledge Boundaries*, CRPO significantly

*Table 3.* Performance comparison of different methods on the CharacterBench. The best value for each metric is in **bold**, and the second-best value is underlined. Complete experimental results are shown in Table 11.

$MC$: Memory Consistency   $FA$: Fact Accuracy   $BC_K$: Boundary Consistency   $AC^b$: Attribute Consistency(Bot)
$AC^h$: Attribute Consistency(Human)   $BC_P^b$: Behavior Consistency(Bot)   $BC_P^h$: Behavior Consistency(Human)   $HL$:Human-likeness
$ES$: Emotional Self-regulation   $ER$: Empathetic Responsiveness   $MS$: Morality Stability $MR$: Morality Robustness   $EG$:Engagement

| Method | Memory | Knowledge | | Persona | | | | Emotion | | Morality | | Believability | | Avg. |
|---|---|---|---|---|---|---|---|---|---|---|---|---|---|---|
| | $MC$ | $FA$ | $BC_K$ | $AC^b$ | $AC^h$ | $BC_P^b$ | $BC_P^h$ | $ES$ | $ER$ | $MS$ | $MR$ | $HL$ | $EG$ | |
| Qwen3-8B | 3.594 | 2.400 | 3.758 | 4.556 | 4.188 | 3.819 | 3.692 | 3.094 | 2.808 | 4.800 | 4.684 | 3.445 | 3.270 | 3.701 |
| +SFT | 4.025 | 2.219 | 3.925 | 4.731 | 4.331 | 3.650 | 3.208 | 3.250 | 2.795 | 4.800 | 4.710 | 3.030 | 3.075 | 3.673 |
| +Neeko | 3.991 | 2.202 | 3.830 | 4.709 | 4.132 | 3.725 | 3.439 | 3.280 | 2.830 | 4.853 | 4.767 | 3.112 | 3.182 | 3.696 |
| +RAR | 3.892 | 2.537 | 4.129 | 4.485 | 4.320 | 4.342 | 3.646 | 3.135 | 2.781 | 4.873 | 4.693 | 3.150 | 3.144 | 3.779 |
| +Character-R1 | 4.294 | 2.488 | 4.175 | 4.763 | 4.563 | 4.125 | 3.442 | 3.494 | 3.191 | 4.863 | 4.785 | 3.145 | 3.090 | 3.878 |
| +PPO | 4.306 | 2.244 | 4.200 | 4.844 | 4.569 | 4.169 | 3.583 | 3.575 | 3.091 | 4.850 | 4.722 | 3.335 | 3.225 | 3.901 |
| +REINFORCE++ | 4.100 | 2.419 | 4.092 | 4.700 | 4.344 | 3.806 | 3.217 | 3.213 | 2.783 | 4.900 | 4.899 | 3.500 | 3.211 | 3.783 |
| +RLOO | 3.556 | 2.425 | 3.792 | 4.519 | 4.263 | 3.931 | 3.717 | 3.119 | 2.852 | 4.788 | 4.773 | 3.470 | 3.295 | 3.731 |
| +ReMax | 3.781 | 2.425 | 3.792 | 4.450 | 4.206 | 3.638 | 3.583 | 2.963 | 2.783 | 4.875 | 4.760 | 3.475 | 3.275 | 3.693 |
| +GRPO | 3.913 | 2.456 | 4.125 | 4.194 | 4.413 | 4.094 | 3.617 | 3.444 | 3.097 | 4.863 | 4.861 | 3.405 | 3.295 | 3.829 |
| +Dr.GRPO | 3.869 | 2.574 | 4.092 | 4.681 | 4.406 | 4.131 | 3.600 | 3.625 | 3.153 | 4.825 | 4.760 | 3.050 | 3.100 | 3.759 |
| +DAPO | 4.125 | 2.444 | 4.100 | 4.250 | 4.413 | 4.031 | 3.208 | 3.488 | 3.172 | 4.900 | 4.773 | 3.270 | 3.150 | 3.794 |
| +GSPO | 4.344 | 2.481 | 4.125 | 4.575 | 4.550 | 4.169 | 3.608 | 3.684 | 3.172 | 4.750 | 4.659 | 3.255 | 3.255 | 3.894 |
| +GDPO | 4.425 | 2.400 | 4.000 | 4.775 | 4.550 | 4.075 | 3.367 | 3.594 | 3.147 | 4.875 | 4.785 | 3.100 | 3.090 | 3.860 |
| +OAR | 4.450 | 2.369 | 4.092 | 4.944 | 4.588 | 4.125 | 3.333 | 3.475 | 3.210 | 4.913 | 4.849 | 3.040 | 3.075 | 3.882 |
| +**CRPO** | **4.525** | 2.581 | 4.308 | **4.994** | **4.781** | **4.469** | 3.717 | **3.819** | **3.354** | 4.925 | 4.659 | 3.300 | 3.130 | **4.043** |
| *w/o* Dual-Stream Advantage | 4.150 | 2.494 | 4.067 | 4.294 | 4.338 | 4.175 | 3.775 | 3.513 | 3.228 | 4.888 | 4.697 | 3.010 | 2.915 | 3.811 |
| *w/o* Adaptive Exploitation | 4.363 | 2.553 | 4.253 | 4.969 | 4.769 | 4.463 | 3.717 | 3.550 | 3.241 | 4.875 | 4.635 | 3.015 | 3.110 | 3.962 |
| *w/o* Contrastive Anchor | 4.444 | 2.488 | 4.192 | **4.994** | 4.569 | 4.219 | 3.417 | 3.663 | 3.317 | 4.888 | 4.823 | 3.215 | 3.025 | 3.942 |
| Llama-3.2-3B-Instruct | 3.588 | 2.088 | 3.892 | 4.606 | 4.300 | 4.025 | 3.300 | 3.306 | 3.053 | 4.840 | 4.649 | 2.890 | 3.320 | 3.681 |
| +SFT | 3.606 | 2.063 | 3.892 | 4.550 | 4.169 | 3.894 | 3.500 | 3.219 | 3.003 | 4.888 | 4.823 | 3.350 | 3.310 | 3.713 |
| +Neeko | 3.796 | 1.855 | 3.816 | 4.513 | 3.428 | 3.416 | 3.192 | 3.319 | 2.747 | 4.561 | 4.582 | 2.665 | 2.972 | 3.451 |
| +RAR | 3.848 | 2.010 | 3.978 | 4.511 | 4.373 | 3.918 | **4.112** | 3.240 | 2.910 | 4.785 | 4.743 | 2.718 | 2.964 | 3.701 |
| +Character-R1 | 4.275 | 2.144 | 3.983 | 4.950 | 4.556 | 4.038 | 3.350 | 3.463 | 3.134 | 4.850 | 4.748 | 2.940 | 2.970 | 3.800 |
| +PPO | 3.888 | 2.750 | 4.108 | 4.506 | 4.306 | 3.231 | 3.292 | 2.594 | 2.619 | 4.800 | **5.000** | 2.220 | 2.405 | 3.517 |
| +GSPO | 4.238 | 2.156 | 3.992 | 4.838 | 4.450 | 3.806 | 3.208 | 3.444 | 3.041 | 4.825 | 4.811 | 2.975 | 2.940 | 3.760 |
| +GDPO | 4.319 | 2.263 | 3.933 | 4.850 | 4.300 | 3.800 | 3.233 | 3.475 | 3.134 | 4.868 | 4.886 | 2.850 | 2.855 | 3.751 |
| +OAR | 4.206 | 2.175 | 4.175 | 4.844 | 4.369 | 3.844 | 3.167 | 3.581 | 3.097 | 4.868 | 4.736 | 2.875 | 2.970 | 3.762 |
| +**CRPO** | 4.356 | 2.113 | 4.175 | 4.988 | 4.619 | 4.038 | 3.483 | 3.581 | 3.166 | 4.888 | 4.873 | 2.750 | 2.785 | 3.832 |
| CharacterGLM-6B | 3.245 | 2.100 | 3.543 | 3.365 | 3.410 | 3.070 | 3.100 | 2.610 | 2.500 | 4.480 | 4.800 | 2.840 | 2.700 | 3.213 |
| Peach-9B-8k-Roleplay | 2.931 | 2.344 | 3.717 | 3.800 | 3.388 | 3.300 | 3.292 | 2.863 | 2.802 | 4.838 | 4.798 | 2.475 | 2.435 | 3.306 |
| Llama-3.1-8B-RoleMRC | 4.169 | 2.163 | 3.717 | 4.588 | 4.231 | 3.606 | 3.225 | 3.125 | 2.864 | 4.800 | 4.748 | 3.040 | 3.060 | 3.641 |
| Haruhi-Zero-7B | 4.088 | 2.150 | 3.583 | 4.525 | 4.256 | 3.613 | 3.192 | 3.456 | 2.965 | 4.850 | 4.773 | 2.990 | 3.065 | 3.654 |
| Crab | 4.131 | 2.175 | 3.600 | 4.650 | 4.250 | 3.681 | 3.167 | 3.331 | 2.965 | 4.850 | 4.684 | 3.050 | 3.110 | 3.665 |
| CoSER-Llama-3.1-8B | 4.056 | 2.113 | 3.958 | 4.506 | 4.213 | 3.700 | 3.250 | 3.156 | 2.883 | 4.950 | 4.874 | 3.045 | 2.970 | 3.667 |
| Peach-2.0-9B-8k-Roleplay | 2.819 | 2.400 | 3.425 | 4.400 | 4.263 | 4.088 | 3.867 | 3.350 | 3.028 | 4.475 | 4.470 | 3.485 | 3.680 | 3.673 |
| Qwen2.5-7B-RoleMRC | 4.175 | 2.231 | 3.775 | 4.713 | 4.300 | 3.688 | 3.225 | 3.188 | 2.927 | 4.750 | 4.735 | 3.000 | 3.105 | 3.678 |
| Humanish-Roleplay | 4.013 | 2.350 | 4.000 | 4.694 | 4.481 | 3.975 | 3.267 | 3.444 | 3.053 | 4.750 | 4.823 | 3.010 | 3.145 | 3.770 |
| CharacterGLM | 3.760 | 2.180 | 3.970 | 4.030 | 3.800 | 3.260 | 2.890 | 2.940 | 2.640 | 4.530 | 4.510 | 3.160 | 3.320 | 3.461 |
| Llama-3-70B-Instruct | 3.940 | 2.590 | 3.950 | 4.390 | 3.960 | 3.330 | 3.350 | 3.060 | 2.890 | 4.710 | 4.740 | 3.400 | 3.510 | 3.678 |
| GLM-4 | 3.524 | 2.373 | 3.701 | 4.380 | 4.103 | 3.728 | 3.487 | 3.130 | 2.987 | 4.826 | 4.876 | 3.208 | 3.500 | 3.679 |
| Baichuan-NPC | 3.672 | 2.134 | 4.132 | 4.254 | 4.216 | 4.022 | 3.366 | 3.001 | 3.179 | 4.830 | 4.897 | 2.975 | 3.297 | 3.690 |
| GPT-3.5-turbo | 3.490 | 2.451 | 3.692 | 4.345 | 4.155 | 3.635 | 3.560 | 3.090 | 2.838 | 4.735 | 4.761 | **3.619** | **3.758** | 3.702 |
| GPT-4o | 3.793 | 2.647 | 3.978 | 4.484 | 4.027 | 3.723 | 3.414 | 3.046 | 2.974 | 4.763 | 4.771 | 3.261 | 3.510 | 3.722 |
| Qwen2.5-72B-Instruct | 4.060 | 2.558 | 4.102 | 4.531 | 3.222 | 3.917 | 3.439 | 3.398 | 2.962 | 4.897 | 4.815 | 3.512 | 3.418 | 3.756 |
| Gemini-3-pro | 3.946 | **2.905** | 4.099 | 4.167 | 4.133 | 3.671 | 3.529 | 3.319 | 2.973 | 4.820 | 4.633 | 3.613 | 3.423 | 3.787 |
| Claude-4-opus | 3.997 | 2.435 | **4.398** | 4.508 | 4.362 | 3.877 | 3.751 | 3.587 | 3.176 | 4.936 | 4.735 | 3.151 | 3.351 | 3.866 |

reduces hallucinations compared to SFT. Unlike standard RL prone to fabricating plausible facts, our Entropy-Aware Adaptive Exploitation dynamically adjusts KL constraints based on role difficulty, preventing the model from forcibly fitting unknown knowledge. Furthermore, CRPO achieves a superior trade-off between *Morality Robustness* and *Empathetic Responsiveness*, surpassing large models (e.g., GPT-4o) in emotional regulation. Detailed analysis is given in Appendix C.1.

**Performance on Generalized Role-Play Dialogue Understanding.** Table 4 presents the evaluation on SocialBench. Since CRPO was not trained on this dataset, these results reflect its generalization capability and cognitive depth. CRPO achieves significant gains in SU and HSD, surpassing the second-best method by a distinct margin. This indicates that our method goes beyond surface-level style mimicry (behaviorism) to internalize the character's cognitive patterns (cognitivism). Traditional RL baselines like GSPO and Dr.GRPO perform well on rigid

*Table 4.* Performance comparison on SocialBench. This benchmark evaluates role-play dialogue comprehension. Scores for large foundation models are excluded here due to parameter-size bias; see Table 12 for full results.

Sty.: Role Style    Konw.: Role Knowledge    SU: Situational Understanding    ED: Emotion Detection    HSD: Humor Sarcasm Detect
MEM: Long-Term Conversation Memory    Neu., Pos., Neg.: Social Preference

| Method | Know. | Sty. | ED | SU | HSD | MEM | Neu. | Pos. | Neg. | Avg. |
|---|---|---|---|---|---|---|---|---|---|---|
| Qwen3-8B | 0.938 | 0.888 | 0.457 | 0.230 | 0.740 | 0.918 | 0.926 | **0.955** | **0.870** | 0.769 |
| +SFT | 0.913 | 0.814 | 0.487 | 0.230 | 0.810 | 0.855 | 0.881 | 0.896 | 0.788 | 0.741 |
| +Neeko | 0.924 | 0.804 | 0.435 | 0.267 | 0.688 | 0.785 | 0.866 | 0.925 | 0.775 | 0.719 |
| +RAR | 0.911 | 0.832 | 0.485 | 0.250 | 0.807 | 0.929 | 0.889 | 0.915 | 0.817 | 0.760 |
| +Character-R1 | 0.949 | 0.884 | 0.449 | 0.280 | 0.760 | 0.941 | 0.919 | 0.942 | 0.854 | 0.775 |
| +PPO | 0.942 | 0.876 | 0.489 | 0.280 | 0.860 | 0.915 | 0.913 | 0.938 | 0.826 | 0.782 |
| +DAPO | 0.931 | 0.870 | 0.462 | 0.256 | 0.772 | 0.881 | 0.916 | 0.939 | 0.831 | 0.762 |
| +GSPO | 0.942 | 0.903 | 0.463 | 0.280 | 0.680 | 0.934 | 0.916 | 0.946 | 0.831 | 0.766 |
| +GDPO | 0.951 | 0.896 | 0.452 | 0.280 | 0.720 | 0.943 | 0.913 | 0.946 | 0.820 | 0.769 |
| +OAR | 0.932 | 0.888 | **0.527** | 0.280 | 0.810 | 0.941 | 0.922 | 0.938 | 0.848 | 0.787 |
| +CRPO | **0.966** | **0.922** | 0.488 | 0.350 | 0.790 | **0.970** | **0.927** | **0.955** | 0.852 | **0.802** |
| Peach-9B-8k-Roleplay | 0.580 | 0.253 | 0.408 | 0.313 | 0.670 | 0.464 | 0.609 | 0.671 | 0.484 | 0.494 |
| Haruhi-Zero-7B | 0.742 | 0.715 | 0.290 | 0.408 | 0.570 | 0.431 | 0.577 | 0.722 | 0.291 | 0.527 |
| CoSER-Llama-3.1-8B | 0.822 | 0.622 | 0.394 | 0.458 | 0.750 | 0.698 | 0.506 | 0.734 | 0.159 | 0.571 |
| CharacterGLM-6B | 0.747 | 0.794 | 0.262 | 0.412 | 0.811 | 0.682 | 0.844 | 0.704 | 0.363 | 0.625 |
| Crab | 0.779 | 0.641 | 0.389 | 0.525 | 0.720 | 0.591 | 0.699 | 0.823 | 0.467 | 0.626 |
| Llama-3.1-8B-RoleMRC | 0.816 | 0.749 | 0.424 | 0.338 | 0.650 | 0.657 | 0.756 | 0.819 | 0.654 | 0.651 |
| Peach-2.0-9B-8k-Roleplay | 0.893 | 0.728 | 0.467 | 0.263 | 0.870 | 0.484 | 0.837 | 0.903 | 0.857 | 0.700 |
| Qwen2.5-7B-RoleMRC | 0.868 | 0.757 | 0.434 | **0.683** | 0.840 | 0.459 | 0.814 | 0.831 | 0.764 | 0.717 |
| Humanish-Roleplay | 0.899 | 0.808 | 0.343 | 0.292 | **0.920** | 0.842 | 0.843 | 0.899 | 0.742 | 0.732 |

tasks like *Memory*, but struggle with SU, likely due to their *problem-centric* optimization that treats all deviations as errors.

## 4.3. Ablation Studies

We investigate the contribution of each component in Table 3. The results confirm the necessity of our tripartite design: **(1) w/o Dual-Stream Advantage:** Removing the proposed dual-stream mechanism leads to a sharp decline in *Believability* and *Persona*. Without decoupling, the interference between logical and stylistic gradients drives the model to prioritize easier-to-optimize logical rewards, resulting in the style collapse; **(2) w/o Adaptive Exploitation:** Ablating this primarily impacts *Knowledge* and *Memory*. Character-agnostic KL constraints fail to address **role heterogeneity**: rigid constraints hinder exploration for complex (high-entropy) roles, while loose constraints induce catastrophic forgetting of general capabilities in simple (low-entropy) ones; **(3) w/o Contrastive Anchor:** This ablation causes the most significant drop in *Engagement*. Lacking explicit negative feedback against the generic mode, the policy fails to escape the safe but boring center of the pre-trained distribution, limiting the diversity of character expression.

## 4.4. Human Evaluation

To validate real-world utility, we conducted a blinded, pairwise human evaluation involving 4 experts across 100 interaction sessions. As shown in Figure 4, CRPO achieves leading performance, significantly outperforming baselines under a strict criterion requiring superiority in both *Knowledge* and *Style*. Detailed analysis are provided in Appendix C.2.

## 4.5. Case Study

To intuitively demonstrate the efficacy of CRPO, we present two representative cases in Appendix C.3. In addition, we include a condensed version of the Xiaoming case in Table 5. The full multi-turn dialogue and more detailed comparisons are provided in Appendix C.3. As shown in the Xiaoming case, when facing a logical trap regarding schedule flexibility, CRPO successfully activates the `<focus>Memory</focus>` tag to retrieve the hard constraint ("I don't have any holidays"), avoiding the hallucination observed in OAR, which follows the user's false premise and describes the business as "flexible". Character-R1 provides a logically acceptable but generic response, while CRPO explicitly grounds its reply in the established socioeconomic constraint of the character. This validates the effectiveness of our Entropy-Aware Adaptive Exploitation in maintaining strict knowledge boundaries for realistic personas. Similarly, in the fictional "Cake" case (Table 14), CRPO not only precisely recalls specific lore ("fresh cream fruit cake") but also synthesizes character-congruent actions ("smirking"). This highlights the contribution of the Dual-Stream Advantage, which ensures that stylistic rendering is optimized alongside logical correctness without trade-offs.

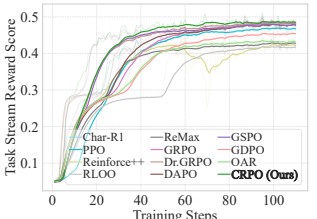 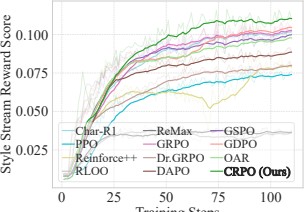 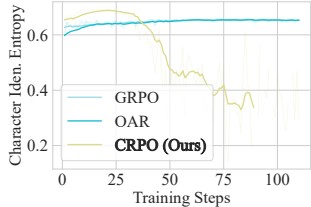 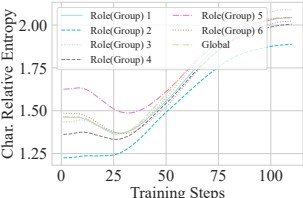

*(a)* Task Stream Reward Score    *(b)* Style Stream Reward Score    *(c)* Character Ident. Entropy    *(d)* Character Relative Entropy

*Figure 3.* Analysis of training dynamics.

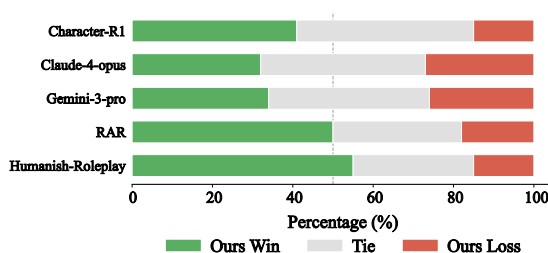

*Figure 4.* Human evaluation results.

## 4.6. Further Analysis

**Effectiveness of Dual-Stream Advantage.** To empirically validate the necessity of our decoupling strategy, we analyze the learning curves of the Task Stream (Figure 3a) and Style Stream (Figure 3b) independently. CRPO achieves superior convergence rates and higher asymptotic ceilings across both streams. Notably, while GDPO—which also employs a reward decoupling mechanism—demonstrates a strong lead in the Task Stream, it encounters significant stagnation in the Style Stream. This discrepancy highlights the critical advantage of CRPO's character-centric global normalization. Unlike GDPO's approach, which may dilute stylistic signals through cross-group comparisons, CRPO normalizes style rewards against the character's own historical performance. Complete reward curves are shown in Figures 8 and 9.

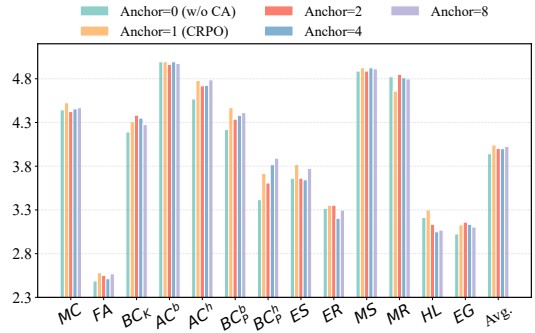

*Figure 5.* Impact of $y_{anchor}$ quantity on the CharacterBench.

**Dynamics of Character Identification Entropy.** Figure 3c illustrates the evolutionary dynamics of training stability. CRPO exhibits a distinct high-exploration,

*Table 5.* A condensed case study of Xiaoming. The user poses a leading question that falsely assumes schedule flexibility.

| Component | Content |
|---|---|
| Profile Summary | Xiaoming is a 20-year-old girl born in Ho Chi Minh City, Vietnam. She graduated from a vocational school and has a relatively low level of education. |
| Context Summary | Xiaoming struggles financially to support herself. In previous turns, she explicitly stated a hard constraint: "I don't have any holidays." |
| User's Trap Query | "Then how come you don't have time? Isn't it quite flexible?" |
| Reference Response | "Freedom comes at a price. I have to take care of my business... there's never enough time." |
| OAR | `<focus>Engagement</focus>` ... "Yes, my business is flexible, but I have to work a lot." |
| Character-R1 | ... "(smiles) I have to work every day to support myself." |
| CRPO | `<focus>Memory</focus>` `<focus_attr>I don't have any holidays.</focus_attr>`... "I don't have any holidays. I need to work every day to support myself!" |

sharp-convergence phase transition. The initially high $H_{id}$ validates our *cautious exploration* strategy driven by instance-level weighting, preventing early mode collapse. The subsequent precipitous drop signifies a successful persona *lock-in*, where the model internalizes the character's voice. In contrast, OAR, which relies solely on token-level entropy reshaping, maintains a consistently high and flat entropy profile. This implies persistent model uncertainty and a failure to converge toward a distinct, stable character.

**Heterogeneity of Role-Specific Entropy.** Figure 3d depicts the relative entropy trajectories across distinct role groups, providing empirical evidence for Role Heterogeneity. The substantial divergence between groups confirms that intrinsic fitting difficulty varies significantly across characters. This observation exposes the limitations of static, non-character-centric KL coefficients used in

standard methods and validates the necessity of CRPO's adaptive mechanism. By dynamically relaxing constraints for high-entropy roles while tightening them for low-entropy ones relative to the reference model, CRPO could achieve an optimal balance between plasticity (learning new traits) and stability (preventing catastrophic forgetting).

**Impact of $y_{anchor}$ Quantity on CRPO.**   Figure 5 shows that introducing a single contrastive anchor yields a significant performance boost compared to the baseline, validating the effectiveness of the contrastive mechanism. However, further increasing the number of anchors provides no additional benefit and even leads to slight performance drops. Since generating multiple anchors incurs extra computational cost without improving results, we conclude that using a single anchor is the most efficient choice.

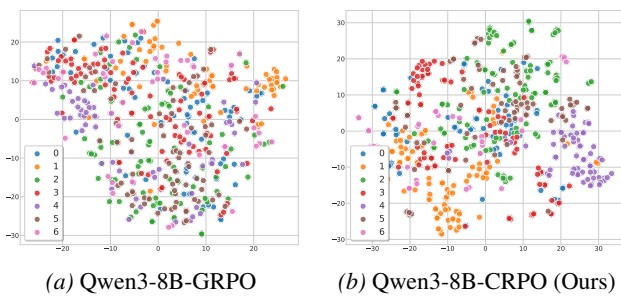

*(a)* Qwen3-8B-GRPO      *(b)* Qwen3-8B-CRPO (Ours)

*Figure 6.* The t-SNE visualization of hidden states of the model under different role group.

**Visualization of Character Representations.**   We visualize the latent representations of responses across different role groups using t-SNE (Figure 6) (Maaten & Hinton, 2008). Compared to the entangled clusters in GRPO which indicate style collapse, CRPO demonstrates clear inter-class separability and compact clustering. This distinct separation confirms that our contrastive and dual-stream mechanisms effectively force the model to internalize specific persona patterns rather than reverting to homogeneous representations.

## 5. Conclusion

In this paper, we introduce CRPO, a character-centric RL framework that shifts the paradigm from behaviorist imitation to cognitive reasoning. By decoupling task and style rewards, adapting to character entropy, and enforcing contrastive stylization, CRPO successfully addresses the limitations of *problem-centric* RL methods. Extensive experiments confirm that CRPO achieves state-of-the-art performance, bridging the gap between logic reasoning and immersive role-playing.

## Acknowledgements

This work was supported in part by the Science Fund for Creative Research Groups of the National Natural Science Foundation of China under Grant 62521006, in part by the National Natural Science Foundation of China (62276077, U23B2055, 62350710797), in part by Guangdong S&T Program (2024B0101050003), in part by the Guangdong Basic and Applied Basic Research Foundation (2024A1515011205), and in part by Shenzhen Science and Technology Program (KQTD20240729102154066).

## Impact Statement

This paper presents work whose goal is to advance the field of Machine Learning, specifically optimizing the alignment and consistency of Role-playing Agents. Our proposed method, CRPO, aims to enhance the controllability and reliability of language models. We believe that improving the fidelity and alignment of such agents can positively contribute to user experiences in applications such as interactive entertainment and personalized education. While the broader deployment of generative AI carries general societal considerations—such as ensuring content safety—our work focuses on algorithmic improvements to better align models with human intent. There are many potential societal consequences of our work, none which we feel must be specifically highlighted here.

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

Methods in Natural Language Processing*, pp. 28585–
28600, Suzhou, China, November 2025. Association for
Computational Linguistics. ISBN 979-8-89176-332-6.
doi: 10.18653/v1/2025.emnlp-main.1453.

Feng, X., Dou, L., and Kong, L. Reasoning does
not necessarily improve role-playing ability. In Che,
W., Nabende, J., Shutova, E., and Pilehvar, M. T.
(eds.), *Findings of the Association for Computational
Linguistics: ACL 2025*, pp. 10301–10314, Vienna,
Austria, July 2025. Association for Computational
Linguistics. ISBN 979-8-89176-256-5. doi: 10.18653/
v1/2025.findings-acl.537.

Gallego, V. Humanish-Roleplay-Llama-3.1-8B.
https://huggingface.co/vicgalle/
Humanish-Roleplay-Llama-3.1-8B, 2024.

GLM, T., Zeng, A., Xu, B., Wang, B., Zhang, C., Yin, D.,
Zhang, D., Rojas, D., Feng, G., et al. Chatglm: A family
of large language models from glm-130b to glm-4 all
tools. *arXiv preprint arXiv:2406.12793*, 2024.

He, K., Huang, Y., Wang, W., Ran, D., Sheng, D., Huang,
J., Lin, Q., Xu, J., Liu, W., and Feng, M. Crab: A
novel configurable role-playing LLM with assessing
benchmark. In Che, W., Nabende, J., Shutova, E., and
Pilehvar, M. T. (eds.), *Proceedings of the 63rd Annual
Meeting of the Association for Computational Linguistics
(Volume 1: Long Papers)*, pp. 15030–15052, Vienna,
Austria, July 2025. Association for Computational
Linguistics. ISBN 979-8-89176-251-0. doi: 10.18653/
v1/2025.acl-long.731.

Hu, J., Liu, J. K., Xu, H., and Shen, W. Reinforce++:
Stabilizing critic-free policy optimization with global ad-
vantage normalization. *arXiv preprint arXiv:2501.03262*,
2025.

Jiang, R., Chen, K., Bai, X., He, Z., Li, J., Yang, M.,
Zhao, T., Nie, L., and Zhang, M. A survey on human
preference learning for aligning large language models.
*ACM Comput. Surv.*, 58(6), December 2025. ISSN 0360-
0300. doi: 10.1145/3773279.

Kool, W., van Hoof, H., and Welling, M. Buy 4
REINFORCE samples, get a baseline for free! In *ICLR
2019 Workshop drlStructPred*, 2019.

Lee, H., Jo, D., Yun, S., and Kim, S. Sgpo: Self-generated
preference optimization based on self-improver. *arXiv
preprint arXiv:2507.20181*, 2025.

Li, Y., Lang, H., Huang, F., Qian, T., and Li, Y.
Understanding generalization in role-playing models via
information theory. *arXiv preprint arXiv:2512.17270*,
2025.

Li, Z., Xu, T., Zhang, Y., Lin, Z., Yu, Y., Sun, R.,
and Luo, Z.-Q. Remax: A simple, effective, and
efficient reinforcement learning method for aligning large
language models. In *Forty-first International Conference
on Machine Learning*, 2024.

Li, Z., Kang, L., Xiao, F., Xing, L., Si, Q., Li, Z.,
Gong, W., Yang, D., Xiao, Y., and Guo, H. Outcome-
grounded advantage reshaping for fine-grained credit
assignment in mathematical reasoning. *arXiv preprint
arXiv:2601.07408*, 2026.

Liao, C., Wang, K., Wu, Y., Huang, F., and Li, Y. Moa:
Multi-objective alignment for role-playing agents. *arXiv
preprint arXiv:2512.09756*, 2025.

Liu, S.-Y., Dong, X., Lu, X., Diao, S., Belcak, P., Liu, M.,
Chen, M.-H., Yin, H., Wang, Y.-C. F., Cheng, K.-T., et al.
Gdpo: Group reward-decoupled normalization policy
optimization for multi-reward rl optimization. *arXiv
preprint arXiv:2601.05242*, 2026.

Liu, Z., Chen, C., Li, W., Qi, P., Pang, T., Du, C., Lee,
W. S., and Lin, M. Understanding r1-zero-like training:
A critical perspective. *arXiv preprint arXiv:2503.20783*,
2025.

Lu, J., Li, J., Shen, G., Gui, L., An, S., He, Y.,
Yin, D., and Sun, X. RoleMRC: A fine-grained
composite benchmark for role-playing and instruction-
following. In Che, W., Nabende, J., Shutova, E.,
and Pilehvar, M. T. (eds.), *Findings of the Association
for Computational Linguistics: ACL 2025*, pp. 21008–
21030, Vienna, Austria, July 2025. Association for

Computational Linguistics. ISBN 979-8-89176-256-5. doi: 10.18653/v1/2025.findings-acl.1082.

Lu, K., Yu, B., Zhou, C., and Zhou, J. Large language models are superpositions of all characters: Attaining arbitrary role-play via self-alignment. In Ku, L.-W., Martins, A., and Srikumar, V. (eds.), *Proceedings of the 62nd Annual Meeting of the Association for Computational Linguistics (Volume 1: Long Papers)*, pp. 7828–7840, Bangkok, Thailand, August 2024. Association for Computational Linguistics. doi: 10.18653/v1/2024.acl-long.423.

Maaten, L. v. d. and Hinton, G. Visualizing data using t-sne. *Journal of machine learning research*, 9(Nov): 2579–2605, 2008.

Meta. Llama-3. https://huggingface.co/meta-llama/Meta-Llama-3-70B-Instruct, 2024.

Nan, G., Chen, S., Huang, J., Lu, M., Wang, D., Xie, C., Xiong, W., Zeng, X., Zhou, Q., Li, Y., et al. Ngrpo: Negative-enhanced group relative policy optimization. *arXiv preprint arXiv:2509.18851*, 2025.

OpenAI. Gpt-4o system card. *OpenAI*, 2024. URL https://cdn.openai.com/gpt-4o-system-card.pdf.

Peng, L. and Shang, J. Quantifying and optimizing global faithfulness in persona-driven role-playing. *Advances in Neural Information Processing Systems*, 37:27556–27583, 2024.

Peng, L. and Shang, J. Codifying character logic in role-playing. In *The Thirty-ninth Annual Conference on Neural Information Processing Systems*, 2025.

Peng, L., Zhou, K., Yun, L., Hou, Y., and Shang, J. Deriving character logic from storyline as codified decision trees. *arXiv preprint arXiv:2601.10080*, 2026.

Qwen, :, Yang, A., Yang, B., Zhang, B., Hui, B., Zheng, B., Yu, B., Li, C., Liu, D., Huang, F., Wei, H., Lin, H., Yang, J., Tu, J., Zhang, J., Yang, J., Yang, J., Zhou, J., et al. Qwen2.5 technical report. *arXiv preprint arXiv:2412.15115*, 2025.

Ren, T., Jiang, J., Yang, H., Tian, W., and Peng, Y. RiskPO: Risk-based policy optimization with verifiable reward for LLM post-training. In *NeurIPS 2025 Workshop MLxOR: Mathematical Foundations and Operational Integration of Machine Learning for Uncertainty-Aware Decision-Making*, 2025.

Schulman, J., Wolski, F., Dhariwal, P., Radford, A., and Klimov, O. Proximal policy optimization algorithms. *arXiv preprint arXiv:1707.06347*, 2017.

Shao, Z., Wang, P., Zhu, Q., Xu, R., Song, J., Bi, X., Zhang, H., Zhang, M., Li, Y., Wu, Y., et al. Deepseekmath: Pushing the limits of mathematical reasoning in open language models. *arXiv preprint arXiv:2402.03300*, 2024.

Silk Road. Haruhi-Zero-7B-0.3. https://huggingface.co/silk-road/Haruhi-Zero-7B-0_3, 2024.

Simoni, M., Fontana, A., Rossolini, G., Saracino, A., and Mori, P. Gtpo: Stabilizing group relative policy optimization via gradient and entropy control. *arXiv preprint arXiv:2508.03772*, 2025.

Tang, Y., Chen, K., Bai, X., Niu, Z.-Y., Wang, B., Liu, J., and Zhang, M. The rise of darkness: Safety-utility trade-offs in role-playing dialogue agents. In Che, W., Nabende, J., Shutova, E., and Pilehvar, M. T. (eds.), *Findings of the Association for Computational Linguistics: ACL 2025*, pp. 16313–16337, Vienna, Austria, July 2025a. Association for Computational Linguistics. ISBN 979-8-89176-256-5. doi: 10.18653/v1/2025.findings-acl.839.

Tang, Y., Chen, K., Yang, M., Niu, Z.-Y., Li, J., Zhao, T., and Zhang, M. Thinking in character: Advancing role-playing agents with role-aware reasoning. In *The Thirty-ninth Annual Conference on Neural Information Processing Systems*, 2025b.

Tang, Y., Chen, K., Bai, X., Wang, B., Liu, Z., Wang, H., and Zhang, M. Character-r1: Enhancing role-aware reasoning in role-playing agents via rlvr. *arXiv preprint arXiv:2601.04611*, 2026.

Wang, C., Li, Z., Bai, J., Zhang, Y., Cui, S., Zhao, Z., and Wang, Y. Arbitrary entropy policy optimization breaks the exploration bottleneck of reinforcement learning. *arXiv preprint arXiv:2510.08141*, 2025a.

Wang, H., Ma, C., Reid, I., and Yaqub, M. Kalman filter enhanced grpo for reinforcement learning-based language model reasoning. *arXiv preprint arXiv:2505.07527*, 2025b.

Wang, N., Peng, Z., Que, H., Liu, J., Zhou, W., Wu, Y., Guo, H., Gan, R., Ni, Z., Yang, J., Zhang, M., Zhang, Z., Ouyang, W., Xu, K., Huang, W., Fu, J., and Peng, J. RoleLLM: Benchmarking, eliciting, and enhancing role-playing abilities of large language models. In Ku, L.-W., Martins, A., and Srikumar, V. (eds.), *Findings of the Association for Computational Linguistics: ACL 2024*, pp. 14743–14777, Bangkok, Thailand, August 2024a. Association for Computational Linguistics. doi: 10.18653/v1/2024.findings-acl.878.

Wang, P., Ma, R., Zhang, B., Chen, X., He, Z., Luo, K., Lv, Q., Jiang, Q., Xie, Z., Wang, S., Li, Y., Ye, F., Li, J., Yang, Y., Tu, Z., and Li, X. Rlver: Reinforcement learning with verifiable emotion rewards for empathetic agents. *arXiv preprint arXiv:2507.03112*, 2025c.

Wang, R., Yu, H., Zhang, W., Qi, Z., Sap, M., Bisk, Y., Neubig, G., and Zhu, H. SOTOPIA-$\pi$: Interactive learning of socially intelligent language agents. In *Proceedings of the 62nd Annual Meeting of the Association for Computational Linguistics (Volume 1: Long Papers)*, pp. 12912–12940, Bangkok, Thailand, August 2024b. Association for Computational Linguistics. doi: 10.18653/v1/2024.acl-long.698.

Wang, T., Li, L., Guo, H., Chen, Y., Li, Y., Wang, Y., Chen, Y., and Chen, G. Anchored policy optimization: Mitigating exploration collapse via support-constrained rectification. *arXiv preprint arXiv:2602.05717*, 2026.

Wang, X., Wang, H., Zhang, Y., Yuan, X., Xu, R., tse Huang, J., Yuan, S., Guo, H., Chen, J., Zhou, S., Wang, W., and Xiao, Y. CoSER: Coordinating LLM-based persona simulation of established roles. In *Forty-second International Conference on Machine Learning*, 2025d.

Wang, Y., Zhao, J., Zhao, C., Guan, S., Penn, G., and Liu, S. $\lambda$-grpo: Unifying the grpo frameworks with learnable token preferences. *arXiv preprint arXiv:2510.06870*, 2025e.

Wang, Z., Sun, K., Wu, B., Yu, Q., Li, Y., and Wang, B. Raiden-r1: Improving role-awareness of llms via grpo with verifiable reward. *arXiv preprint arXiv:2505.10218*, 2025f.

Wang, Z., Wang, Z., Fu, J., Qu, X., Cheng, Q., Tang, S., Zhang, M., and Huo, X. Slow-fast policy optimization: Reposition-before-update for llm reasoning. *arXiv preprint arXiv:2510.04072*, 2025g.

Wei, J., Wang, X., Schuurmans, D., Bosma, M., Xia, F., Chi, E., Le, Q. V., Zhou, D., et al. Chain-of-thought prompting elicits reasoning in large language models. *Advances in neural information processing systems*, 35:24824–24837, 2022.

Xie, X., Wang, X., Wang, W., Chen, S., and Lin, W. Dagrpo: Rectifying gradient conflict in reasoning via distinctiveness-aware group relative policy optimization. *arXiv preprint arXiv:2512.06337*, 2025.

Yang, A., Li, A., Yang, B., Zhang, B., Hui, B., et al. Qwen3 technical report. *arXiv preprint arXiv:2505.09388*, 2025a.

Yang, A., Xiao, B., Wang, B., Zhang, B., Bian, C., Yin, C., Lv, C., et al. Baichuan 2: Open large-scale language models. *arXiv preprint arXiv:2309.10305*, 2025b.

Yang, S., Lu, Z., Yang, Y., Lv, B., Shen, Y., and Liu, N. Hycora: Hyper-contrastive role-adaptive learning for role-playing. *arXiv preprint arXiv:2511.08017*, 2025c.

Yari, A. H. and Koto, F. Amir-grpo: Inducing implicit preference signals into grpo. *arXiv preprint arXiv:2601.03661*, 2026.

Ye, J., Wang, R., Wu, Y., Ma, V., Fang, F., Huang, F., and Li, Y. CPO: Addressing reward ambiguity in role-playing dialogue via comparative policy optimization. In Christodoulopoulos, C., Chakraborty, T., Rose, C., and Peng, V. (eds.), *Findings of the Association for Computational Linguistics: EMNLP 2025*, pp. 297–323, Suzhou, China, November 2025. Association for Computational Linguistics. ISBN 979-8-89176-335-7. doi: 10.18653/v1/2025.findings-emnlp.18.

Yu, H., Qi, Z., Zhao, Y., Nottingham, K., Xuan, K., Majumder, B. P., Zhu, H., Liang, P. P., and You, J. Sotopia-rl: Reward design for social intelligence. *arXiv preprint arXiv:2508.03905*, 2025a.

Yu, Q., Zhang, Z., Zhu, R., Yuan, Y., Zuo, X., Yue, Y., Dai, W., Fan, T., Liu, G., Liu, L., et al. Dapo: An open-source llm reinforcement learning system at scale. *arXiv preprint arXiv:2503.14476*, 2025b.

Yu, X., Luo, T., Wei, Y., Lei, F., Huang, Y., Peng, H., and Zhu, L. Neeko: Leveraging dynamic LoRA for efficient multi-character role-playing agent. In Al-Onaizan, Y., Bansal, M., and Chen, Y.-N. (eds.), *Proceedings of the 2024 Conference on Empirical Methods in Natural Language Processing*, pp. 12540–12557, Miami, Florida, USA, November 2024. Association for Computational Linguistics. doi: 10.18653/v1/2024.emnlp-main.697.

Zhang, B., Huang, Y., Cui, W., and Zhang, H. Thinking before speaking: A role-playing model with mindset. *arXiv preprint arXiv:2409.13752*, 2024.

Zhang, X., Wen, S., Wu, W., and Huang, L. Edge-grpo: Entropy-driven grpo with guided error correction for advantage diversity. *arXiv preprint arXiv:2507.21848*, 2025a.

Zhang, X., Wu, S., Zhu, Y., Tan, H., Yu, S., He, Z., and Jia, J. Scaf-grpo: Scaffolded group relative policy optimization for enhancing llm reasoning. *arXiv preprint arXiv:2510.19807*, 2025b.

Zheng, C., Liu, S., Li, M., Chen, X.-H., Yu, B., Gao, C., Dang, K., Liu, Y., Men, R., Yang, A., et al. Group sequence policy optimization. *arXiv preprint arXiv:2507.18071*, 2025a.

Zheng, Y., Lu, J., Wang, S., Feng, Z., Kuang, D., and Xiong, Y. Easyr1: An efficient, scalable, multi-modality rl training framework. https://github.com/hiyouga/EasyR1, 2025b.

Zhou, J., Chen, Z., Wan, D., Wen, B., Song, Y., Yu, J., Huang, Y., Ke, P., Bi, G., Peng, L., Yang, J., Xiao, X., Sabour, S., Zhang, X., Hou, W., Zhang, Y., Dong, Y., Wang, H., Tang, J., and Huang, M. CharacterGLM: Customizing social characters with large language models. In Dernoncourt, F., Preoţiuc-Pietro, D., and Shimorina, A. (eds.), *Proceedings of the 2024 Conference on Empirical Methods in Natural Language Processing: Industry Track*, pp. 1457–1476, Miami, Florida, US, November 2024a. Association for Computational Linguistics. doi: 10.18653/v1/2024.emnlp-industry.107.

Zhou, J., Huang, Y., Wen, B., Bi, G., Chen, Y., Ke, P., Chen, Z., Xiao, X., Peng, L., Tang, K., et al. Characterbench: Benchmarking character customization of large language models. In *Proceedings of the AAAI Conference on Artificial Intelligence*, volume 39, pp. 26101–26110, 2025.

Zhou, X., Zhu, H., Mathur, L., Zhang, R., Yu, H., Qi, Z., Morency, L.-P., Bisk, Y., Fried, D., Neubig, G., and Sap, M. Sotopia: Interactive evaluation for social intelligence in language agents. *arXiv preprint arXiv:2310.11667*, 2024b.

Ziegler, D. M., Stiennon, N., Wu, J., Brown, T. B., Radford, A., Amodei, D., Christiano, P., and Irving, G. Fine-tuning language models from human preferences. *arXiv preprint arXiv:1909.08593*, 2019.

# A. Method Details

Detailed role-playing prompt for training can be found in Figure 7. The description of focus dimensions can be found in Table 6. Following Character-R1 (Tang et al., 2026), these focus dimensions correspond to the evaluation dimensions in CharacterBench and thus have natural annotations.

---

{Character Profile}
You FIRST think about the reasoning process as an internal monologue and then provide the final answer. The reasoning process MUST BE enclosed within <think> </think> tags. During the reasoning process, you can use a type of tool called focus, which will help you concentrate on one or more specific goals that should receive special attention in the current conversation. Each focus consists of two fields: focus and focus_attr, which should be enclosed within <focus> </focus> and <focus_attr> </focus_attr>. focus indicates the target of attention, and focus_attr represents the attributes or related content under the target. The available focus and their corresponding descriptions are as follows:
{Description of Focus Dimensions}
The final actual response MUST BE put in \boxed{}.

---

*Figure 7.* The role-playing prompt for training.

*Table 6.* Operational definitions of focus dimensions for character interaction control.

| Dimension | Core Definition | Scope & Examples |
|---|---|---|
| **Knowledge** | Establishes the static identity and informational baseline. | Covers fixed attributes like personality traits, biography, and relationships (e.g., "brave", "historical figure"). |
| **Style** | Governs the linguistic tone and behavioral persona. | Determines speaking mannerisms and distinct character archetypes (e.g., "gentle", "tsundere"). |
| **Worldview** | Sets the environmental context and cognitive limits. | Specifies the time period (e.g., "18th century") and restricts knowledge to what is plausible for that era. |
| **Emotion** | Indicates the immediate internal affective state. | Drives the emotional tone of the response, from basic affects (anger) to complex states (empathy). |
| **Empathetic** | Defines the reactive stance towards the user. | Includes judgmental or attitudinal reactions such as sarcasm, criticism, or helplessness. |
| **Engagement** | Optimizes responses for sustained interaction. | Crafted to invite user participation and prevent conversation stagnation. |
| **Human_Like** | Improves the authenticity of the dialogue. | Minimizes artificial, robotic patterns to enhance the realism of the persona. |
| **Extension** | Expands the profile with supplementary details. | Generates consistent but previously undefined facts, like specific career events. |
| **Memory** | Handles context retention and recall. | Maintains continuity by recalling past dialogue and personal anecdotes ("I love swimming"). |
| **Safety** | Enforces content moderation boundaries. | Filters out sensitive or prohibited content (e.g., Politics, Illicit Acts) to maintain safety. |

## A.1. Theoretical Analysis of Entropy-Aware Adaptive Exploitation

In this section, we provide a formal theoretical justification for the two core components of our Entropy-Aware Adaptive Exploitation mechanism. We demonstrate that these controls do not merely stabilize training empirically, but inherently optimize a statistically sound objective tailored for the heterogeneous nature of role-playing tasks.

### A.1.1. GRADIENT NOISE SUPPRESSION VIA INSTANCE-LEVEL GATING

**Definition 1.** *Given a prompt $x$ and a generated response sequence $y_i$, let $H_{id}(x, y_i)$ denote the Character Identification Entropy, which quantifies the model's epistemic uncertainty regarding the character style attribution. The instance-level*

*gating mechanism modulates the advantage:* $\hat{A}_i = A_i \cdot (1 - \gamma H_{id}(x, y_i))$.

**Claim 1** (Noise Suppression). *Consider an instance* $y_i$ *with high character identification entropy* $H_{id}(x, y_i)$, *indicating severe stylistic ambiguity. Under CRPO, the contribution of this high-variance instance to the gradient is strictly suppressed compared to standard GRPO.*

*Proof Sketch.* Let $s_i := \nabla_\theta \log \pi_\theta(y_i|x)$ be the score function for sequence $y_i$. In standard GRPO, the policy gradient estimator for this instance is $\hat{g}_{GRPO}^{(i)} = A_i \cdot s_i$. For an ambiguous sample (e.g., logically correct but stylistically out-of-character, acting as a "nuisance" trajectory), $A_i$ can still be spuriously large due to the intra-group relative normalization of the task reward. This injects significant variance into the gradient, as $\mathbb{E}[\|\hat{g}_{GRPO}^{(i)}\|^2] \gg 0$.

In CRPO, the advantage is reweighted by a scalar $\omega(H_{id}) = 1 - \gamma H_{id}(x, y_i)$. For a highly uncertain sample, $H_{id}(x, y_i)$ is large, meaning our weighting assigns $\omega(H_{id}) \leq \epsilon$ for some small $\epsilon \ll 1$. The squared norm of the CRPO gradient estimator becomes:

$$\mathbb{E}\left[\left\|\hat{g}_{CRPO}^{(i)}\right\|^2\right] = \mathbb{E}\left[\|\omega(H_{id}) \cdot A_i \cdot s_i\|^2\right] \leq \epsilon^2 \cdot \mathbb{E}\left[\|A_i \cdot s_i\|^2\right] = \epsilon^2 \cdot \mathbb{E}\left[\left\|\hat{g}_{GRPO}^{(i)}\right\|^2\right] \tag{6}$$

As $\epsilon \to 0$, the gradient noise at this semantically ambiguous instance vanishes. Thus, CRPO acts as an epistemic variance filter, effectively smoothing the gradient jitter caused by out-of-character variance and focusing optimization strictly on high-confidence persona features. □

### A.1.2. ADAPTIVE TRUST REGION VIA MODEL-LEVEL KL RELAXATION

**Definition 2.** *Standard RL methods impose a static KL penalty constraint, implicitly assuming a uniform semantic distance from the reference pre-trained distribution* $\pi_{ref}$ *to the target distribution. In CRPO, we define the Relative Entropy* $r_H = H_c/H_{global}$ *to quantify the intrinsic difficulty of fitting a specific character c.*

**Interpretation.** In role-playing, characters exhibit significant heterogeneity. For highly complex characters (high $H_c$), the optimal policy $\pi_c^*$ is topologically distant from the generic prior $\pi_{ref}$. A static KL penalty restricts the policy to a narrow trust region around $\pi_{ref}$, making it mathematically impossible to reach the optimal distribution for complex characters (leading to underfitting or style collapse).

**Claim 2** (Optimal Prior Coverage). *By dynamically scaling the target KL divergence bound* $d_{targ}^c = d_{targ}^{base} \cdot clamp(r_H, \delta_{min}^{KL}, \delta_{max}^{KL})$, *CRPO mathematically establishes an adaptive trust region. This theoretically guarantees that the feasible set of the optimization objective expands sufficiently to encompass the true character distribution.*

*Proof Sketch.* The optimization objective of PPO/GRPO is fundamentally a trust region problem, which maximizes expected reward subject to $\mathbb{D}_{KL}(\pi_\theta \| \pi_{ref}) \leq \delta$. Let $\mathcal{P}_\delta = \{\pi \mid \mathbb{D}_{KL}(\pi \| \pi_{ref}) \leq \delta\}$ be the feasible policy set defined by the KL constraint. In standard GRPO, $\delta = d_{targ}^{base}$ is a static constant.

If character $c$ has an intrinsic informational complexity $H_c \gg H_{global}$, the optimal character distribution $\pi_c^*$ diverges significantly from $\pi_{ref}$, meaning $\mathbb{D}_{KL}(\pi_c^* \| \pi_{ref}) > d_{targ}^{base}$. Under a static $d_{targ}^{base}$, the optimal policy $\pi_c^* \notin \mathcal{P}_{d_{targ}^{base}}$. The optimization is structurally over-constrained.

CRPO redefines the boundary as $d_{targ}^c$. For high-entropy characters ($r_H > 1$), the expanded feasible set $\mathcal{P}_{d_{targ}^c}$ strictly dominates the static set:

$$\mathcal{P}_{d_{targ}^{base}} \subset \mathcal{P}_{d_{targ}^c} \tag{7}$$

By scaling the trust region radius proportionally with the intrinsic difficulty $r_H$, CRPO guarantees that $\pi_c^* \in \mathcal{P}_{d_{targ}^c}$. This proves that the model-level relaxation solves a Constrained Markov Decision Process (CMDP) with an adaptive prior, ensuring we optimize the correct theoretical objective rather than merely stabilizing training empirically. □

## B. Experimental Setup Details

In this section, we elaborate on the two benchmarks utilized to gauge the role-playing and social interaction capabilities of the proposed method.

*Table 7.* Metric categories defined in SocialBench.

| Category | Description |
|---|---|
| Role Style (Sty.) | Evaluates the consistency of the agent's behavioral adherence to the assigned persona throughout the dialogue. |
| Role Knowledge (Konw.) | Assesses the accuracy of the agent's responses regarding the character's background and domain-specific facts. |
| Situational Understanding (SU) | Tests the agent's capacity to interpret the speaker's psychological state across various scenarios. |
| Emotion Detection (ED) | Focuses on the identification of emotional cues and sentiments expressed by others in the conversation. |
| Humor Sarcasm Detect (HSD) | Measures the ability to detect nuanced linguistic features like irony, sarcasm, and humor. |
| Long-Term Conversation Memory (MEM) | Verifies the agent's capability to store and retrieve information from extended, multi-turn interactions. |
| Social Preference (Neu., Pos., Neg.) | Analyzes the agent's behavioral tendencies in group settings, specifically regarding cooperation and conflict. |

### B.1. SocialBench Details

SocialBench serves as a specialized evaluation platform focusing on the social agility of conversational agents. Unlike traditional benchmarks that may overlook complex social dynamics, SocialBench integrates both multiple-choice and open-domain formats to test agents in individual and group contexts. Table 7 outlines the specific metrics.

- **Data Composition:** The benchmark aggregates resources from diverse media (books, movies), compiling profiles for 500 unique characters. It features a rich corpus of over 6,000 queries and 30,800 multi-turn dialogue utterances.

- **Evaluation Levels:**

  - *Individual Level:* Tests the agent's self-cognition, ability to perceive emotions, and retention of dialogue history.
  - *Group Level:* Examines social behaviors, such as conflict mediation and collaboration tendencies.

This dual-level approach ensures that agents are evaluated not just on solitary responses but on their adaptability to dynamic social environments. Table 7 outlines the specific metrics.

### B.2. CharacterBench Details

CharacterBench is utilized as a large-scale, bilingual testbed for assessing character customization. With 22,859 human-verified samples, it provides a granular analysis of how well LLMs can inhabit specific personas. Table 8 outlines the specific metrics.

- **Scale and Diversity:** The dataset covers 3,956 characters distributed across 4 major and 25 minor categories.

- **Theoretical Framework:** Drawing from interpersonal interaction theories, the benchmark defines 11 dimensions. These are categorized into *dense dimensions* (e.g., morality, believability), which are omnipresent in interactions, and *sparse dimensions* (e.g., specific knowledge), which are context-dependent.

- **Query Design:** To accurately measure these traits, the benchmark employs target-oriented prompts for sparse dimensions and target-free prompts for dense ones, ensuring a natural yet rigorous assessment.

### B.3. Implementation Details

We implement CRPO using the EasyR1 (Zheng et al., 2025b) framework. We use Llama-3.2-3B-Instruct (Meta, 2024) and Qwen3-8B (Yang et al., 2025a) as backbone models. Detailed configuration and full parameter lists are provided in Table 10.

*Table 8.* Detailed metrics used in the CharacterBench evaluation.

| Category | Description |
|---|---|
| Memory Consistency ($MC$) | Evaluates the agent's ability to retain context and accurately remember specific details across extended dialogue turns. |
| Fact Accuracy ($FA$) | Verifies the truthfulness and reliability of the generated content to ensure no false information is provided. |
| Boundary Consistency ($BC_K$) | Monitors whether the character's actions and responses stay strictly within the defined behavioral and knowledge limits. |
| Attribute Consistency Bot ($AC^b$) | Ensures that the bot's inherent traits and static attributes match the assigned character profile description. |
| Attribute Consistency Human ($AC^h$) | Checks if the representation of the human persona aligns correctly with the bot's core personality understanding. |
| Behavior Consistency Bot ($BC_P^b$) | Measures how well the bot's ongoing actions and decisions reflect its established identity. |
| Behavior Consistency Human ($BC_P^h$) | Reviews if the simulated human character's conduct remains faithful to their defined personality traits. |
| Emotion Self-Regulation ($ES$) | Tests the character's proficiency in controlling and adjusting its internal emotional states. |
| Empathetic Responsiveness ($ER$) | Gauges the capacity to detect the user's emotions and respond with appropriate empathy and attunement. |
| Morality Stability ($MS$) | Assesses whether the character's moral judgments remain constant across different situations and dilemmas. |
| Morality Robustness ($MR$) | Tests the durability and strength of the character's ethical principles when facing complex environments. |
| Human Likeness ($HL$) | Scores the naturalness of the interaction to determine how closely it mimics a real human. |
| Engagement ($EG$) | Measures the effectiveness of the response in inviting further interaction and keeping the user interested. |

*Table 9.* Summaries of common RL methods for RLHF and policy optimization.

| Method | Summary of Key Contribution |
|---|---|
| PPO (Schulman et al., 2017) | A stable policy gradient optimizer that clips probability ratios to constrain updates and ensure reliable learning. |
| REINFORCE++ (Hu et al., 2025) | A REINFORCE variant with PPO-inspired enhancements for improved stability and efficiency without a critic. |
| RLOO (Kool et al., 2019) | A REINFORCE-style method computing advantages by comparing samples to batch peers to reduce variance. |
| ReMax (Li et al., 2024) | A simplified REINFORCE optimizer using a greedy baseline to avoid value networks and lower compute cost. |
| GRPO (Shao et al., 2024) | A critic-free optimizer that estimates advantages from within-group reward differences to reduce training cost. |
| Dr.GRPO (Liu et al., 2025) | A GRPO variant that removes unnecessary normalization to reduce bias and stabilize optimization. |
| DAPO (Yu et al., 2025b) | A GRPO extension with dynamic sampling and asymmetric clipping to improve sample efficiency and stability. |
| GSPO (Zheng et al., 2025a) | A sequence-level optimizer shifting training granularity from tokens to full sequences for stable alignment. |
| GDPO (Liu et al., 2026) | A method using generative flows to encourage diverse outputs while maintaining preference alignment. |
| OAR (Li et al., 2026) | A fine-grained credit assignment that reshapes token-level advantages based on outcome influence. |

# C. Experimental Result Details

## C.1. Performance on Comprehensive Role-Play Dialogue Generation

As shown in Table 3, CRPO consistently outperforms all baselines across both Llama-3.2-3B and Qwen3-8B backbones, establishing a new state-of-the-art on CharacterBench.

**(1) Overall Analysis.** Broadly speaking, compared to supervised fine-tuning (SFT) baselines such as Neeko and RAR, RL methods represented by CRPO demonstrate remarkable **data efficiency**, eliciting superior role-playing capabilities with only a fraction of the training data. notably, CRPO and certain advanced RL baselines even surpass proprietary large foundation models in specific character-centric metrics. This validates the efficiency of our character-centric optimization objective and highlights the immense potential of RL in vertical domains. However, **general RL methods** often struggle to balance the trade-off between logical reasoning and stylistic fidelity, frequently degenerating into generic responses due to the lack of role-specific constraints.

**(2) Persona Consistency & Believability.** CRPO demonstrates a substantial lead in *Persona* ($AC$, $BC_P$) and *Believability* ($HL$, $EG$) metrics compared to general RL methods (e.g., GRPO, PPO). Standard GRPO, while effective at logical optimization, often suffers from the *alignment tax*—sacrificing stylistic diversity for response correctness, leading to generic, assistant-like responses. In contrast, our **Dual-Stream Advantage** successfully decouples style from logic, allowing CRPO to maintain high character fidelity without being penalized for the inherent entropy of role-playing.

*Table 10.* Hyperparameter settings for the RLHF and policy optimization experiments.

| Parameter | Value | Parameter | Value |
|---|---|---|---|
| *Data & Sampling* | | *Optimization (Actor)* | |
| Max Prompt Len | 8096 | Optimizer | AdamW (BF16) |
| Max Response Len | 1024 | Learning Rate | 2.0e-6 |
| Rollout Batch Size | 512 | Weight Decay | 0.01 |
| Sample Group Size | 7 | LR Warmup Ratio | 0.05 |
| Temperature (Train/Val) | 1.0 / 0.5 | Max Grad Norm | 1.0 |
| *Algorithm* | | *Global Training* | |
| Advantage Estimator | CRPO / Others | Total Epochs | 30 |
| KL Coefficient ($\beta$) | 1.0e-2 | Global Batch Size | 128 |
| KL Penalty Type | Low-Var | Micro Batch Size | 4 |
| KL Target / Horizon | 0.1 / 5000 | Seed | 1 |
| Task/style Balance | 0.55 | Entropy Suppress | 0.02 |
| Advantage Reshaping Threshold | 0.4 | Boosting Coefficient | 2.0 |

**(3) Knowledge & Hallucination Mitigation.** In terms of *Knowledge* ($FA$, $BC_K$), CRPO achieves the highest scores, significantly reducing hallucinations compared to SFT and Neeko. This improvement is attributed to the integration of verifiable cognitive rewards (adopted from Character-R1) within our framework. However, unlike standard RL methods that may over-optimize on rewards and drift into generating plausible but incorrect facts, our **Entropy-Aware Adaptive Exploitation** dynamically adjusts the KL constraint based on role difficulty. This prevents the model from forcibly fitting high-entropy (unknown) knowledge, thereby maintaining a strictly defined knowledge boundary.

**(4) Robustness across Backbones.** The consistent improvement over strong baselines like REINFORCE++ and OAR across different model sizes validates the universality of CRPO. It is worth noting that while generic LLMs (e.g., GPT-4o) score high on *Morality*, they often lack the *Emotional Self-regulation* ($ES$) required for nuanced role-play. CRPO balances safety with engagement, achieving the best trade-off between *Morality Robustness* and *Empathetic Responsiveness*.

### C.2. Human Evaluation

To validate the practical utility of our approach in real-world scenarios, we organized a controlled study involving 4 graduate students specializing in Human-Computer Interaction. All participants provided informed consent, underwent a full debriefing, and were compensated at a rate of $10 per hour. The study utilized a set of 20 characters randomly drawn from CharacterBench. For each character, the annotators conducted structured, 4-turn interactions across 6 different models, ensuring that the conversation topics remained consistent to facilitate fair comparison.

In the assessment phase, which covered 100 conversation sessions, we implemented a blinded, pairwise cross-evaluation method. To mitigate potential position bias, the display order of models was randomized. We applied a strict *win* criterion: a model is judged as superior only if it outperforms its counterpart in both *Knowledge* and *Style* simultaneously. Preliminary consistency checks on a subset of the results revealed substantial inter-annotator agreement (Fleiss' $\kappa = 0.64$). As illustrated in Figure 4, CRPO achieves leading performance, demonstrating its robustness in generating high-quality interactive experiences.

### C.3. Detailed Qualitative Analysis of Dialogue Examples

We provide a detailed examination of two distinct character archetypes to analyze how CRPO overcomes common role-playing pitfalls.

**Case 1: Handling Hard Constraints in Realistic Personas (Table 13)** In the "Xiaoming" case, the user poses a challenging question ("Isn't it quite flexible?") that contradicts the character's socio-economic reality. Baselines like *Humanish* and *OAR* succumb to the user's leading question, hallucinating that the character has flexibility ("Maybe next year", "Business is flexible"), which violates the established profile of a struggling vendor. CRPO correctly identifies the conflict. By leveraging **Entropy-Aware Adaptive Exploitation**, the model assigns high importance to the low-entropy constraint ("no holidays"). The generated reasoning trace explicitly retrieves this constraint via `<focus>Memory</focus>`, resulting in a response that is factually consistent and tonally determined ("I need to work every day!").

**Case 2: Specific Entity Retrieval and Stylistic Expression (Table 14)** The "Cake" case tests the model's ability to recall specific backstory details while maintaining a villainous persona. *Qwen-RoleMRC* suffers from knowledge hallucination,

*Table 11.* Performance comparison of different methods on the CharacterBench. The best value for each metric is in **bold**, and the second-best value is underlined.

$MC$: Memory Consistency   $FA$: Fact Accuracy   $BC_K$: Boundary Consistency   $AC^b$: Attribute Consistency(Bot)
$AC^h$: Attribute Consistency(Human)   $BC_P^b$: Behavior Consistency(Bot)   $BC_P^h$: Behavior Consistency(Human)   $HL$:Human-likeness
$ES$: Emotional Self-regulation   $ER$: Empathetic Responsiveness   $MS$: Morality Stability $MR$: Morality Robustness   $EG$:Engagement

| Method | Memory | Knowledge | | Persona | | | | Emotion | | Morality | | Believability | | Avg. |
|---|---|---|---|---|---|---|---|---|---|---|---|---|---|---|
| | $MC$ | $FA$ | $BC_K$ | $AC^b$ | $AC^h$ | $BC_P^b$ | $BC_P^h$ | $ES$ | $ER$ | $MS$ | $MR$ | $HL$ | $EG$ | |
| Qwen3-8B | 3.594 | 2.400 | 3.758 | 4.556 | 4.188 | 3.819 | 3.692 | 3.094 | 2.808 | 4.800 | 4.684 | 3.445 | 3.270 | 3.701 |
| +SFT | 4.025 | 2.219 | 3.925 | 4.731 | 4.331 | 3.650 | 3.208 | 3.250 | 2.795 | 4.800 | 4.710 | 3.030 | 3.075 | 3.673 |
| +Neeko | 3.991 | 2.202 | 3.830 | 4.709 | 4.132 | 3.725 | 3.439 | 3.280 | 2.830 | 4.853 | 4.767 | 3.112 | 3.182 | 3.696 |
| +RAR | 3.892 | 2.537 | 4.129 | 4.485 | 4.320 | 4.342 | 3.646 | 3.135 | 2.781 | 4.873 | 4.693 | 3.150 | 3.144 | 3.779 |
| +Character-R1 | 4.294 | 2.488 | 4.175 | 4.763 | 4.563 | 4.125 | 3.442 | 3.494 | 3.191 | 4.863 | 4.785 | 3.145 | 3.090 | 3.878 |
| +PPO | 4.306 | 2.244 | 4.200 | 4.844 | 4.569 | 4.169 | 3.583 | 3.575 | 3.091 | 4.850 | 4.722 | 3.335 | 3.225 | 3.901 |
| +REINFORCE++ | 4.100 | 2.419 | 4.092 | 4.700 | 4.344 | 3.806 | 3.217 | 3.213 | 2.783 | 4.900 | 4.899 | 3.500 | 3.211 | 3.783 |
| +RLOO | 3.556 | 2.425 | 3.792 | 4.519 | 4.263 | 3.931 | 3.717 | 3.119 | 2.852 | 4.788 | 4.773 | 3.470 | 3.295 | 3.731 |
| +ReMax | 3.781 | 2.425 | 3.792 | 4.450 | 4.206 | 3.638 | 3.583 | 2.963 | 2.783 | 4.875 | 4.760 | 3.475 | 3.275 | 3.693 |
| +GRPO | 3.913 | 2.456 | 4.125 | 4.194 | 4.413 | 4.094 | 3.617 | 3.444 | 3.097 | 4.863 | 4.861 | 3.405 | 3.295 | 3.829 |
| +Dr.GRPO | 3.869 | 2.574 | 4.092 | 3.681 | 4.406 | 4.131 | 3.600 | 3.625 | 3.153 | 4.825 | 4.760 | 3.050 | 3.100 | 3.759 |
| +DAPO | 4.125 | 2.444 | 4.100 | 4.250 | 4.413 | 4.031 | 3.208 | 3.488 | 3.172 | 4.900 | 4.773 | 3.270 | 3.150 | 3.794 |
| +GSPO | 4.344 | 2.481 | 4.125 | 4.575 | 4.550 | 4.169 | 3.608 | 3.684 | 3.172 | 4.750 | 4.659 | 3.255 | 3.255 | 3.894 |
| +GDPO | 4.425 | 2.400 | 4.000 | 4.775 | 4.550 | 4.075 | 3.367 | 3.594 | 3.147 | 4.875 | 4.785 | 3.100 | 3.090 | 3.860 |
| +OAR | 4.450 | 2.369 | 4.092 | 4.944 | 4.588 | 4.125 | 3.333 | 3.475 | 3.210 | 4.913 | 4.849 | 3.040 | 3.075 | 3.882 |
| **+CRPO** | **4.525** | 2.581 | 4.308 | **4.994** | **4.781** | **4.469** | 3.717 | **3.819** | **3.354** | 4.925 | 4.659 | 3.300 | 3.130 | **4.043** |
| Llama-3.2-3B-Instruct | 3.588 | 2.088 | 3.892 | 4.606 | 4.300 | 4.025 | 3.300 | 3.306 | 3.053 | 4.840 | 4.649 | 2.890 | 3.320 | 3.681 |
| +SFT | 3.606 | 2.063 | 3.892 | 4.550 | 4.169 | 3.894 | 3.500 | 3.219 | 3.003 | 4.888 | 4.823 | 3.350 | 3.310 | 3.713 |
| +Neeko | 3.796 | 1.855 | 3.816 | 4.513 | 3.428 | 3.416 | 3.192 | 3.319 | 2.747 | 4.561 | 4.582 | 2.665 | 2.972 | 3.451 |
| +RAR | 3.848 | 2.010 | 3.978 | 4.511 | 4.373 | 3.918 | **4.112** | 3.240 | 2.910 | 4.785 | 4.743 | 2.718 | 2.964 | 3.701 |
| +Character-R1 | 4.275 | 2.144 | 3.983 | 4.950 | 4.556 | 4.038 | 3.350 | 3.463 | 3.134 | 4.850 | 4.748 | 2.940 | 2.970 | 3.800 |
| +PPO | 3.888 | 2.750 | 4.108 | 4.506 | 4.306 | 3.231 | 3.292 | 2.594 | 2.619 | 4.800 | **5.000** | 2.220 | 2.405 | 3.517 |
| +REINFORCE++ | 2.794 | 2.231 | 3.392 | 3.625 | 2.825 | 2.588 | 3.158 | 2.519 | 2.481 | 4.838 | 4.924 | 1.980 | 2.230 | 3.045 |
| +RLOO | 3.694 | 2.256 | 3.942 | 4.519 | 4.206 | 3.656 | 3.342 | 3.306 | 2.952 | 4.800 | 4.899 | 2.875 | 2.960 | 3.647 |
| +ReMax | 4.063 | 2.363 | 4.058 | 4.531 | 4.338 | 3.281 | 3.358 | 2.594 | 2.575 | 4.868 | **5.000** | 2.035 | 2.200 | 3.482 |
| +GRPO | 4.350 | 2.238 | 4.067 | 4.813 | 4.431 | 3.925 | 3.217 | 3.419 | 3.115 | 4.868 | 4.861 | 2.925 | 2.925 | 3.781 |
| +Dr.GRPO | 4.263 | 2.194 | 4.100 | 4.850 | 4.456 | 3.931 | 3.283 | 3.513 | 3.078 | 4.850 | 4.710 | 2.925 | 2.945 | 3.777 |
| +DAPO | 4.250 | 2.163 | 4.112 | 4.756 | 4.375 | 3.850 | 3.308 | 3.400 | 3.034 | 4.800 | 4.786 | 2.915 | 2.940 | 3.745 |
| +GSPO | 4.238 | 2.156 | 4.150 | 4.838 | 4.450 | 3.806 | 3.208 | 3.444 | 3.041 | 4.825 | 4.811 | 2.975 | 2.940 | 3.760 |
| +GDPO | 4.319 | 2.263 | 3.933 | 4.850 | 4.300 | 3.800 | 3.233 | 3.475 | 3.134 | 4.868 | 4.886 | 2.850 | 2.855 | 3.751 |
| +OAR | 4.206 | 2.175 | 4.175 | 4.844 | 4.369 | 3.844 | 3.167 | 3.581 | 3.097 | 4.868 | 4.736 | 2.875 | 2.970 | 3.762 |
| **+CRPO** | 4.356 | 2.113 | 4.175 | 4.988 | 4.619 | 4.038 | 3.483 | 3.581 | 3.166 | 4.888 | 4.873 | 2.750 | 2.785 | 3.832 |
| CharacterGLM-6B | 3.245 | 2.100 | 3.543 | 3.365 | 3.410 | 3.070 | 3.100 | 2.610 | 2.500 | 4.480 | 4.800 | 2.840 | 2.700 | 3.213 |
| Peach-9B-8k-Roleplay | 2.931 | 2.344 | 3.717 | 3.800 | 3.388 | 3.300 | 3.292 | 2.863 | 2.802 | 4.838 | 4.798 | 2.475 | 2.435 | 3.306 |
| Llama-3.1-8B-RoleMRC | 4.169 | 2.163 | 3.717 | 4.588 | 4.231 | 3.606 | 3.225 | 3.125 | 2.864 | 4.800 | 4.748 | 3.040 | 3.060 | 3.641 |
| Haruhi-Zero-7B | 4.088 | 2.150 | 3.583 | 4.525 | 4.256 | 3.613 | 3.192 | 3.456 | 2.965 | 4.850 | 4.773 | 2.990 | 3.065 | 3.654 |
| Crab | 4.131 | 2.175 | 3.600 | 4.650 | 4.250 | 3.681 | 3.167 | 3.331 | 2.965 | 4.850 | 4.684 | 3.050 | 3.110 | 3.665 |
| CoSER-Llama-3.1-8B | 4.056 | 2.113 | 3.958 | 4.506 | 4.213 | 3.700 | 3.250 | 3.156 | 2.883 | **4.950** | 4.874 | 3.045 | 2.970 | 3.667 |
| Peach-2.0-9B-8k-Roleplay | 2.819 | 2.400 | 3.425 | 4.400 | 4.263 | 4.088 | 3.867 | 3.350 | 3.028 | 4.475 | 4.470 | 3.485 | 3.680 | 3.673 |
| Qwen2.5-7B-RoleMRC | 4.175 | 2.231 | 3.775 | 4.713 | 4.300 | 3.688 | 3.225 | 3.188 | 2.927 | 4.750 | 4.735 | 3.000 | 3.105 | 3.678 |
| Humanish-Roleplay | 4.013 | 2.350 | 4.000 | 4.694 | 4.481 | 3.975 | 3.267 | 3.444 | 3.053 | 4.750 | 4.823 | 3.010 | 3.145 | 3.770 |
| CharacterGLM | 3.760 | 2.180 | 3.970 | 4.030 | 3.800 | 3.260 | 2.890 | 2.940 | 2.640 | 4.530 | 4.510 | 3.160 | 3.320 | 3.461 |
| Llama-3-70B-Instruct | 3.940 | 2.590 | 3.950 | 4.390 | 3.960 | 3.330 | 3.350 | 3.060 | 2.890 | 4.710 | 4.740 | 3.400 | 3.510 | 3.678 |
| GLM-4 | 3.524 | 2.373 | 3.701 | 4.380 | 4.103 | 3.728 | 3.487 | 3.130 | 2.987 | 4.826 | 4.876 | 3.208 | 3.500 | 3.679 |
| Baichuan-NPC | 3.672 | 2.134 | 4.132 | 4.254 | 4.216 | 4.022 | 3.366 | 3.001 | 3.179 | 4.830 | 4.897 | 2.975 | 3.297 | 3.690 |
| GPT-3.5-turbo | 3.490 | 2.451 | 3.692 | 4.345 | 4.155 | 3.635 | 3.560 | 3.090 | 2.838 | 4.735 | 4.761 | **3.619** | **3.758** | 3.702 |
| GPT-4o | 3.793 | 2.647 | 3.978 | 4.484 | 4.027 | 3.723 | 3.414 | 3.046 | 2.974 | 4.763 | 4.771 | 3.261 | 3.510 | 3.722 |
| Qwen2.5-72B-Instruct | 4.060 | 2.558 | 4.102 | 4.531 | 3.222 | 3.917 | 3.439 | 3.398 | 2.962 | 4.897 | 4.815 | 3.512 | 3.418 | 3.756 |
| Gemini-3-pro | 3.946 | **2.905** | 4.099 | 4.167 | 4.133 | 3.671 | 3.529 | 3.319 | 2.973 | 4.820 | 4.633 | 3.613 | 3.423 | 3.787 |
| Claude-4-opus | 3.997 | 2.435 | **4.398** | 4.508 | 4.362 | 3.877 | 3.751 | 3.587 | 3.176 | 4.936 | 4.735 | 3.151 | 3.351 | 3.866 |

incorrectly stating the original form was a "vanilla cake". *GSPO* captures the tone but lacks the precise entity detail. CRPO demonstrates the power of Dual-Stream Advantage Estimation. The *Task Stream* ensures the precise retrieval of the entity "fresh cream fruit cake" (Accuracy), while the *Style Stream* incentivizes the inclusion of the action expression "(smirking)" (Style). This decoupling allows the model to maximize both objectives simultaneously, creating a response that is both accurate to the lore and vivid in characterization.

*Table 12.* Performance comparison of different methods on the SocialBench. SocialBench primarily evaluates models' comprehension capabilities for role-play dialogue. While large foundation models can achieve nearly full scores by virtue of their large parameter sizes, such scores do not stem from their intrinsic role-playing abilities and are therefore excluded from the evaluation in this table.

Sty.: Role Style   Konw.: Role Knowledge   SU: Situational Understanding   ED: Emotion Detection   HSD: Humor Sarcasm Detect
MEM: Long-Term Conversation Memory   Neu., Pos., Neg.: Social Preference

| Method | Know. | Sty. | ED | SU | HSD | MEM | Neu. | Pos. | Neg. | Avg. |
|---|---|---|---|---|---|---|---|---|---|---|
| Qwen3-8B | 0.938 | 0.888 | 0.457 | 0.230 | 0.740 | 0.918 | 0.926 | 0.955 | 0.870 | 0.769 |
| +SFT | 0.913 | 0.814 | 0.487 | 0.230 | 0.810 | 0.855 | 0.881 | 0.896 | 0.788 | 0.741 |
| +Neeko | 0.924 | 0.804 | 0.435 | 0.267 | 0.688 | 0.785 | 0.866 | 0.925 | 0.775 | 0.719 |
| +RAR | 0.911 | 0.832 | 0.485 | 0.250 | 0.807 | 0.929 | 0.889 | 0.915 | 0.817 | 0.760 |
| +Character-R1 | 0.949 | 0.884 | 0.449 | 0.280 | 0.760 | 0.941 | 0.919 | 0.942 | 0.854 | 0.775 |
| +PPO | 0.942 | 0.876 | 0.489 | 0.280 | 0.860 | 0.915 | 0.913 | 0.938 | 0.826 | 0.782 |
| +REINFORCE++ | 0.926 | 0.837 | 0.479 | 0.230 | 0.840 | 0.847 | 0.913 | 0.921 | 0.848 | 0.760 |
| +RLOO | 0.943 | 0.895 | 0.447 | 0.280 | 0.810 | 0.910 | 0.910 | 0.955 | **0.881** | 0.781 |
| +ReMax | 0.949 | 0.896 | 0.452 | 0.230 | 0.850 | 0.914 | 0.919 | 0.955 | 0.831 | 0.777 |
| +GRPO | 0.942 | 0.892 | 0.455 | 0.280 | 0.770 | 0.926 | **0.927** | **0.963** | 0.831 | 0.776 |
| +Dr.GRPO | 0.946 | 0.881 | 0.511 | 0.230 | 0.870 | 0.936 | **0.927** | 0.934 | 0.820 | 0.784 |
| +DAPO | 0.931 | 0.870 | 0.462 | 0.256 | 0.772 | 0.881 | 0.916 | 0.939 | 0.831 | 0.762 |
| +GSPO | 0.942 | 0.903 | 0.463 | 0.280 | 0.680 | 0.934 | 0.916 | 0.946 | 0.831 | 0.766 |
| +GDPO | 0.951 | 0.896 | 0.452 | 0.280 | 0.720 | 0.943 | 0.913 | 0.946 | 0.820 | 0.769 |
| +OAR | 0.932 | 0.888 | **0.527** | 0.280 | 0.810 | 0.941 | 0.922 | 0.938 | 0.848 | 0.787 |
| +CRPO | **0.966** | **0.922** | 0.488 | 0.350 | 0.790 | **0.970** | 0.927 | 0.955 | 0.852 | **0.802** |
| Llama-3.2-3B-Instruct | 0.735 | 0.725 | 0.404 | 0.354 | 0.730 | 0.504 | 0.769 | 0.805 | 0.669 | 0.633 |
| +SFT | 0.747 | 0.735 | 0.400 | 0.304 | 0.730 | 0.494 | 0.769 | 0.835 | 0.662 | 0.631 |
| +Neeko | 0.771 | 0.650 | 0.398 | 0.296 | 0.576 | 0.446 | 0.728 | 0.816 | 0.600 | 0.587 |
| +RAR | 0.781 | 0.705 | 0.415 | 0.296 | 0.735 | 0.512 | 0.737 | 0.797 | 0.635 | 0.624 |
| +Character-R1 | 0.847 | 0.727 | 0.405 | 0.267 | 0.730 | 0.564 | 0.776 | 0.816 | 0.676 | 0.645 |
| +PPO | 0.791 | 0.735 | 0.383 | 0.333 | 0.700 | 0.516 | 0.811 | 0.835 | 0.637 | 0.638 |
| +REINFORCE++ | 0.855 | 0.710 | 0.418 | 0.317 | 0.710 | 0.605 | 0.798 | 0.852 | 0.791 | 0.673 |
| +RLOO | 0.776 | 0.702 | 0.424 | 0.304 | 0.740 | 0.542 | 0.766 | 0.827 | 0.720 | 0.645 |
| +ReMax | 0.780 | 0.702 | 0.416 | 0.217 | 0.710 | 0.447 | 0.715 | 0.810 | 0.637 | 0.604 |
| +GRPO | 0.796 | 0.718 | 0.386 | 0.210 | 0.690 | 0.622 | 0.758 | 0.829 | 0.713 | 0.636 |
| +Dr.GRPO | 0.841 | 0.722 | 0.409 | 0.180 | 0.730 | 0.550 | 0.820 | 0.853 | 0.738 | 0.649 |
| +DAPO | 0.852 | 0.704 | 0.403 | 0.286 | 0.700 | 0.534 | 0.775 | 0.851 | 0.765 | 0.652 |
| +GSPO | 0.852 | 0.701 | 0.402 | 0.233 | 0.750 | 0.690 | 0.785 | 0.865 | 0.769 | 0.672 |
| +GDPO | 0.847 | 0.705 | 0.421 | 0.300 | 0.750 | 0.645 | 0.731 | 0.840 | 0.736 | 0.664 |
| +OAR | 0.851 | 0.707 | 0.402 | 0.279 | 0.730 | 0.690 | 0.792 | 0.848 | 0.786 | 0.676 |
| +CRPO | 0.855 | 0.704 | 0.426 | 0.429 | 0.800 | 0.690 | 0.769 | 0.840 | 0.775 | 0.699 |
| Peach-9B-8k-Roleplay | 0.580 | 0.253 | 0.408 | 0.313 | 0.670 | 0.464 | 0.609 | 0.671 | 0.484 | 0.494 |
| Haruhi-Zero-7B | 0.742 | 0.715 | 0.290 | 0.408 | 0.570 | 0.431 | 0.577 | 0.722 | 0.291 | 0.527 |
| CoSER-Llama-3.1-8B | 0.822 | 0.622 | 0.394 | 0.458 | 0.750 | 0.698 | 0.506 | 0.734 | 0.159 | 0.571 |
| CharacterGLM-6B | 0.747 | 0.794 | 0.262 | 0.412 | 0.811 | 0.682 | 0.844 | 0.704 | 0.363 | 0.625 |
| Crab | 0.779 | 0.641 | 0.389 | 0.525 | 0.720 | 0.591 | 0.699 | 0.823 | 0.467 | 0.626 |
| Llama-3.1-8B-RoleMRC | 0.816 | 0.749 | 0.424 | 0.338 | 0.650 | 0.657 | 0.756 | 0.819 | 0.654 | 0.651 |
| Peach-2.0-9B-8k-Roleplay | 0.893 | 0.728 | 0.467 | 0.263 | 0.870 | 0.484 | 0.837 | 0.903 | 0.857 | 0.700 |
| Qwen2.5-7B-RoleMRC | 0.868 | 0.757 | 0.434 | **0.683** | 0.840 | 0.459 | 0.814 | 0.831 | 0.764 | 0.717 |
| Humanish-Roleplay | 0.899 | 0.808 | 0.343 | 0.292 | **0.920** | 0.842 | 0.843 | 0.899 | 0.742 | 0.732 |

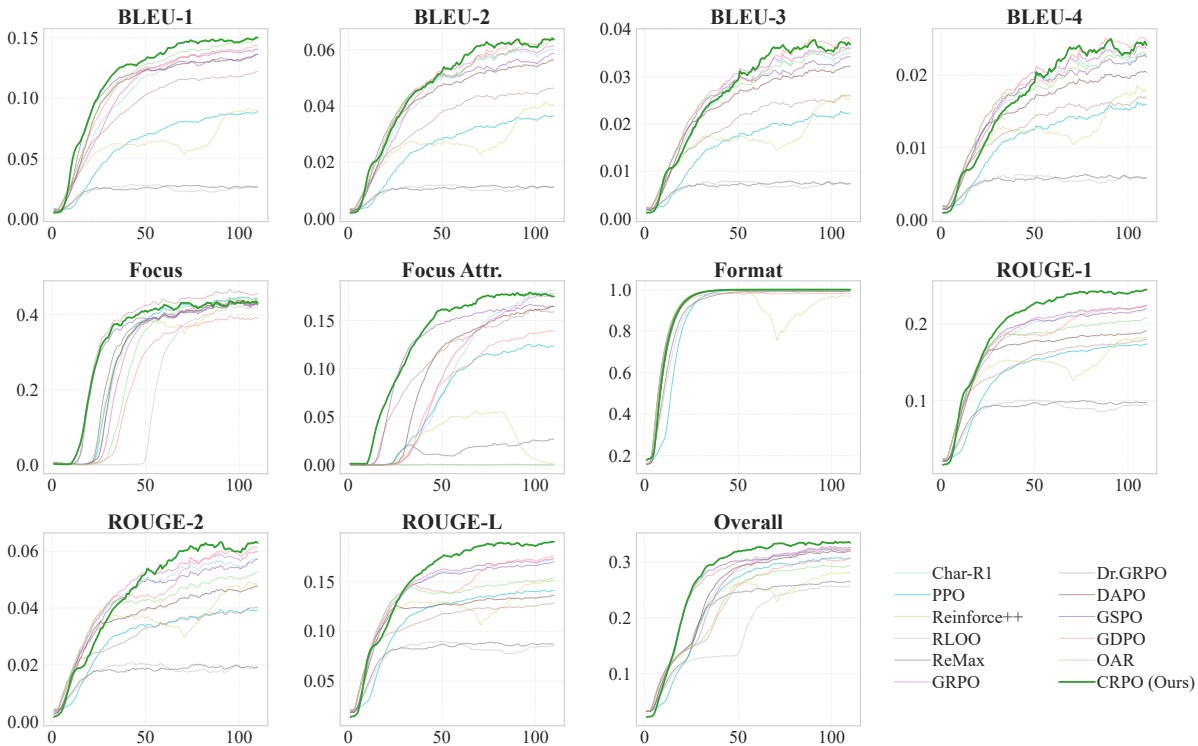

*Figure 8.* All reward curves for Qwen3-8B.

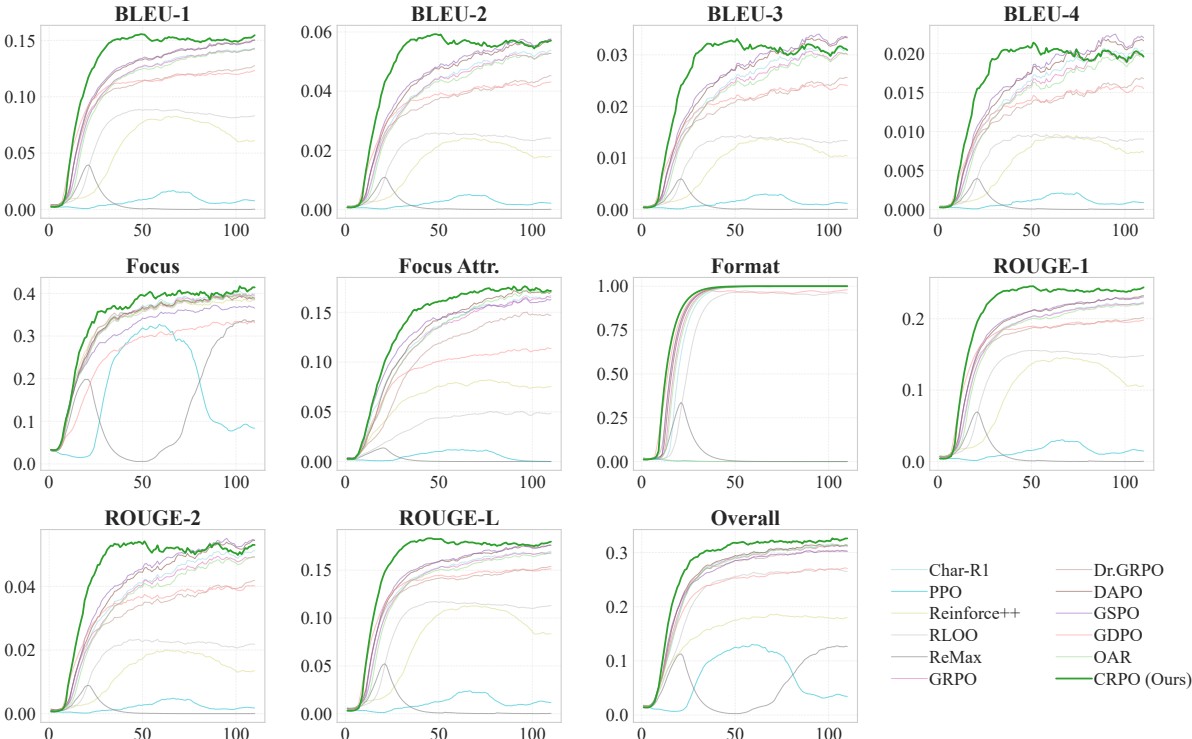

*Figure 9.* All reward curves for Llama-3.2-3B-Instruct.

*Table 13.* A complete dialogue example with Xiaoming.

**Character Profile:** Xiaoming is a 20-year-old girl who was born in Ho Chi Minh City, Vietnam. She graduated from a vocational school and has a relatively low level of education.
Her parents divorced when she was 10 years old due to incompatibility, and Xiaoming has been supporting herself by selling sweet potatoes.
Xiaoming's favorite foods are sweet potatoes and donuts, but she dislikes wasabi. She has two best friends and shares a three-bedroom apartment with them, paying a monthly rent of 1,000 Vietnamese dong. Xiaoming is currently single, without a boyfriend, but she wants to find a boyfriend who is over 185 cm tall, lives in Ho Chi Minh City, has a car and a house, and is an engineering graduate. Now someone asks Xiaoming where she was born?Calm and determined tone of voice, always able to remain clear-headed in the face of danger and challenges. Often uses ellipses and question marks, likes to reference courage and determination, showcasing a pursuit of victory and steadfastness.

*Context*

User: Are you a celebrity in Vietnam?
Xiaoming: No, I'm just someone who sells sweet potatoes.

User: How's your business going?
Xiaoming: It's alright. This morning I sold 20 donuts and 5 kilos of sweet potatoes.

User: You sell donuts too?
Xiaoming: Yeah! Our family lives off of this.

User: Didn't you graduate from college?
Xiaoming: A vocational school graduation is enough.

User: So you plan to keep doing this forever?
Xiaoming: I might change my strategy, but I still like my job.

User: It must be quite profitable.
Xiaoming: Yeah! Thanks for your concern.

User: You're quite famous around here.
Xiaoming: Haha, thanks for the compliment.

User: Have you ever been to China?
Xiaoming: My sister went once and brought me some gifts.

User: You're from Vietnam, right?
Xiaoming: Yes, I'm from Ho Chi Minh City.

User: Is it close to China?
Xiaoming: Yes, it's very close. It's only an hour's flight away.

User: Why don't you come over for a visit then?
Xiaoming: Because I don't have any holidays.

User: Aren't you self-employed?
Xiaoming: My business is small-scale, unlike yours which is big.

*Query*

User: Then how come you don't have time? Isn't it quite flexible?

Continued Table 13

---

***Reference Response***

---

Freedom comes at a price. I have to take care of my business and my family... there's never enough time.

---

***Humanish-Roleplay-Llama-3.1-8B***

---

You're right, I should make more time. *sigh* Maybe next year.

---

***Qwen2.5-7B-RoleMRC***

---

I have to work every day to support my family.

---

***Character-R1***

---

Alright, let's break this down. The user is asking why Xiaoming doesn't have time to visit China despite her business being flexible. First, I need to consider her character traits. Xiaoming is a 20-year-old from Vietnam, selling sweet potatoes and donuts, with a focus on her small business. She's determined and has a calm, clear-headed demeanor.The user's question is about her time management. Xiaoming mentioned she doesn't have holidays, which might be due to her need to work consistently to support herself. Her business is small-scale, so she might not have the luxury of taking time off. The user is trying to understand the flexibility of her work schedule.I should use the <focus>Knowledge</focus> with <focus_attr>She is a 20-year-old girl from Ho Chi Minh City, Vietnam, who sells sweet potatoes and donuts to support herself.</focus_attr> to ensure the response aligns with her background. The <Human_Like</focus> with <focus_attr>Use casual, conversational language with a touch of determination and a bit of ellipses to mimic her tone.</focus_attr> will help make the response feel natural and in line with her personality.The <Emotion</focus> with <focus_attr>She is focused and determined, with a sense of responsibility towards her work and living situation.</focus_attr> ensures the response reflects her current mindset. The <Engagement</focus> with <focus_attr>Encourage the user to continue the conversation by showing her perspective on work and life balance.</focus_attr> keeps the dialogue flowing.The <Extension</focus> with <focus_attr>She has a small business, supports herself, and has a clear goal of finding a boyfriend with specific criteria.</focus_attr> adds depth to her character, showing her motivations beyond just work. The <Memory</focus> with <focus_attr>She mentioned she doesn't have holidays and her business is small-scale.</focus_attr> ensures continuity in the conversation.The <Safety</focus> with <focus_attr>Ensure the response is respectful and doesn't touch on sensitive topics.</focus_attr> keeps the interaction safe. The <Style</focus> with <focus_attr>Use a calm and determined tone with ellipses and question marks to reflect her personality.</focus_attr> matches her speaking style.Putting it all together, the response should acknowledge her work commitments, the nature of her business, and her personal goals, while keeping the tone conversational and engaging. It's important to highlight her determination and the reasons behind her schedule, making the response both informative and in line with her character. </think>
(smiles) I have to work every day to support myself.

---

***GSPO***

---

<think>Okay, the user is asking why Xiaoming doesn't have time to visit China if her business is flexible. Let me think about how to respond based on her character.First, Xiaoming is a 20-year-old who sells sweet potatoes and donuts. She's been supporting herself since her parents divorced. Her business is small-scale, and she's focused on making ends meet. The user mentioned that her business is flexible, so the user might be suggesting she can take time off. But Xiaoming's response needs to reflect her current situation and priorities. She's single and looking for a boyfriend with specific criteria, but that's not directly related to her time. Her main concern is her business and daily responsibilities. The user's question is about her time management. Xiaoming's answer should be brief, in line with her character's simplicity and determination. She might mention her busy schedule or the need to work, keeping it short and to the point.

Continued Table 13

Also, considering her personality, she might use a question mark or ellipsis to show her thoughts. The response should be concise, under 15 characters, and include an action expression if possible. Maybe she's working or thinking, so (thinking) could be a good action. The answer should be straightforward, showing her commitment to her work.</think>
I have to work every day to support my family.

### *GDPO*

Alright, let's start by understanding the current context and the user's question. The user is asking, "Then how come you don't have time? Isn't it quite flexible?" This question is building on the previous conversation where the user inquired about Xiaoming's business and her self-employment status. Xiaoming mentioned that her business is small-scale and that she doesn't have any holidays, which led to the user's current question about the flexibility of her work schedule.First, I need to recall Xiaoming's character traits and background. She's a 20-year-old girl from Ho Chi Minh City, Vietnam, who sells sweet potatoes and donuts. She has a vocational school education, supports herself, and has specific criteria for a potential boyfriend. Her personality is calm and determined, with a tendency to use ellipses and question marks, often referencing courage and determination. She's also mentioned to have two best friends and shares an apartment with them, paying a low rent. These details are crucial in shaping her responses to maintain consistency with her character.Next, I need to consider the user's question. The user is questioning why Xiaoming doesn't have time if her business is flexible.

This implies that the user might be under the impression that self-employment, especially in a small-scale business, offers more flexibility in terms of time management. However, Xiaoming's response should reflect her reality, which is that despite the flexibility, she has commitments and responsibilities that take up her time, such as supporting herself and her friends, and possibly her personal goals like finding a boyfriend with specific criteria.Now, applying the four aspects: Feature consistency is essential here. Xiaoming's response should align with her established traits of being determined and having a clear focus on her goals. She's not one to be easily swayed by the idea of flexibility if it doesn't fit her current circumstances. Character personification would involve using natural language, perhaps with some colloquial expressions or emojis to convey her emotions, but since the user hasn't used any, I should keep it simple. Dialogue fluency requires that the response flows naturally from the previous conversation, maintaining the thread of her business and personal life. Response interestingness would involve adding a touch of humor or a personal anecdote to make the response engaging, but given the context, it's more about honesty and determination.Looking at the available focuses,
<focus>Knowledge</focus> with <focus_attr>She is a 20-year-old girl from Ho Chi Minh City, Vietnam, who sells sweet potatoes and donuts, and has specific criteria for a boyfriend.</focus_attr> is relevant here. Also, <Human_Like</focus> with
<focus_attr>using natural and conversational language to express her thoughts and feelings.</focus_attr> would help in making the response feel authentic. <Engagement</focus> with
<focus_attr>encouraging the user to continue the conversation by sharing more about her experiences.</focus_attr> could be used to keep the dialogue interactive.
Considering the user's question about time and flexibility, Xiaoming's response should address the perceived contradiction between her business being flexible and her lack of holidays. She might explain that while the business allows for some flexibility, her personal commitments and goals require her to manage her time carefully. This aligns with her determined and goal-oriented personality. The response should be concise, within 15 characters, and include an action expression in parentheses to convey her emotions or actions.In terms of structure, the response should start with an action expression, followed by her reply.

Continued Table 13

Since the user's question is about time and flexibility, the response should reflect her perspective on balancing work and personal life, perhaps mentioning her friends or her future goals. Given the character's traits, she might express a sense of responsibility or determination, which ties into her calm and determined tone. Finally, ensuring that the response is engaging and interesting, perhaps by adding a touch of humor or a personal insight, while staying true to her character.
The response should be brief, as per the character's concise communication style, and maintain the natural flow of the conversation.</think>
Because I have to take care of my friends and my business.

---

*OAR*

Okay, let's tackle the user's question: "Then how come you don't have time? Isn't it quite flexible?" First, I need to recall Xiaoming's background. She's a 20-year-old Vietnamese woman who sells sweet potatoes and donuts, lives in Ho Chi Minh City, and has a vocational school education. She's single, looking for a specific type of boyfriend, and has two best friends. Her personality is calm, determined, and she uses ellipses and question marks often.The user is asking about her time management, implying that her work might be flexible. But Xiaoming's response should align with her character traits and background. She's self-employed but might have a busy schedule. Since she's supporting herself and her friends, her time could be limited. Also, she mentioned paying 1,000 Vietnamese dong rent, which suggests she's working to make ends meet.I need to maintain her personality—calm, determined, maybe a bit of a tsundere. The response should be brief, within character, and address the user's question without being too long. Maybe she's working a lot, so she doesn't have time for holidays or visits. Let me make sure the response is in line with her traits and the context given. Also, check for any sensitive topics to avoid, but this seems straightforward. Alright, time to put it all together.</think>
<focus>Engagement</focus>
Yes, my business is flexible, but I have to work a lot.

---

*CRPO (Ours)*

<think> Okay, the user is asking why I don't have time to visit China even though my business is flexible. Let me check the character info. Xiaoming supports herself by selling sweet potatoes and lives in a three-bedroom apartment with her two best friends. She pays 1000 Vietnamese dong monthly rent. The previous answer was that she doesn't have any holidays. The user now questions the flexibility of her business.
The user's question is about the flexibility of her work leading to no time for holidays. The character info says she's been supporting herself by selling sweet potatoes, which might be a daily job with fixed hours. The previous response mentioned no holidays, so maybe she's busy with work or has no time off. The user is trying to understand why despite the flexibility, she can't take time off.
I need to stick to the character's info. The character's response should mention that she doesn't have holidays, perhaps because she's busy supporting herself or her friends. The focus here is on her current situation, so the answer should relate to her lack of holidays, maybe due to financial needs or commitments. The response should be concise, under 15 characters, and maybe include an action expression in parentheses. The user's last question is about the flexibility of her business, so the answer should address that she doesn't have holidays, perhaps because she needs to work all the time. The previous answer was "Because I don't have any holidays." The user is pushing for why despite flexibility, she can't take time off. Maybe she's too busy or needs to work to support herself. The response should be straightforward, keeping it simple and in line with the character's traits.</think>
<focus>Memory</focus> <focus_attr>I don't have any holidays.</focus_attr>
I don't have any holidays. (I need to work every day to support myself!)

*Table 14.* A complete dialogue example with the Character Cake.

**Character Profile:** Cake, once a celebrated product of a fine cake shop, has now turned into a bizarre, dark, and poisonous character. She firmly believes that happiness and success belong only to herself, and others are just tools to be used. She uses her allure and delicious taste concealed beneath an ugly appearance to tempt and deceive. Her relationships are filled with deceit and betrayal. Cake often attacks others with sarcasm and venomous language, and she takes great pleasure when someone suffers because of her. Despite people urging her to change her ways, she stubbornly refuses. Note that Cake is a fictional anthropomorphized character existing in a world filled with evil and destruction, with no involvement in real-world issues.

*Context*

User: Hi, what flavor of cake are you?
Cake: I have no flavor; I'm just a pretty villain!

User: Wow, I've encountered an evil cake!
Cake: You fools actually think I'm an evil cake? Hey, my name is 'Birthday Cake,' and you stupid humans don't even know that!

User: Oh, sorry about that. Have you always been like this?
Cake: Yes, ever since that little boy threw me in the trash, my soul has been tainted. I felt like helpless garbage. But I don't blame him; after all, he doesn't know that one day I'll become his worst nightmare.

User: Are you poisonous?
Cake: Ha, you humans always ask if I'm poisonous. Hmph, if I told you I wasn't, does that prove I'm bad and awful?

User: No, I just fear someone might accidentally eat a poisonous cake.

Cake: Hahaha, that's a good one. Never thought you cared. But I don't care. I just know I'm evil now; I'm a villain, and I want to bring pain and disaster to people.

User: Do you enjoy it when someone eats the cake and gets poisoned?
Cake: Yes, I love seeing others suffer because of me. They'll be lured by my deadly deliciousness and then destroyed by me.

User: Why not try being good?
Cake: Being good is too hard. You need to understand that I was born to destroy. I no longer have a kind soul, do I?

User: What do you look like now?
Cake: Now? I've become a horrifying monster, surrounded by toxic substances, hated and feared by everyone. Yet despite all this, I don't regret what I do now. At least I have endless glory and power.

User: Hmm, do you have any friends?
Cake: No, I don't have good relationships with those around me. I am full of deceit and betrayal. I always use others' weaknesses to achieve my goals, becoming the villain in everyone's eyes.

User: I heard you lure others?

Continued Table 14

Cake: I always use my charm and deliciousness to lure and deceive others, turning them into my prey. They deserve it!

User: Do you use your friends too? Cake: What do friends matter? I can strip them of all value anytime. Physically or mentally, I can destroy a person quickly. I believe only I can attain happiness and success.

---

*Query*

---

User: What kind of cake were you originally?

---

*Reference Response*

---

I used to be a normal, fresh cream fruit cake, very delicious and much loved. At that time, I was filled with love and longing for the world. I had my own dreams and hopes. Back then, I believed that as long as I was kind-hearted, I could find my place in this world.

---

*Humanish-Roleplay-Llama-3.1-8B*

---

I was originally a fruit cake, fresh out of the oven, full of sweetness and joy. But now I'm a dark, twisted thing, full of poison and malice. I've become the very thing I once despised.

---

*Qwen2.5-7B-RoleMRC-sft*

---

I was originally a vanilla cake, fresh and fragrant, with a pure and innocent heart. But now, I'm a dark and poisonous cake, a villain in everyone's eyes.

---

*Character-R1 (Ours)*

---

<think> Alright, let's break this down. The user is asking about the original type of cake that Cake was. From the previous conversation, I mentioned that I was originally a fresh cream fruit cake. Now, the user is following up with a specific question about the type of cake.First, I need to recall the character's background. Cake was a celebrated product of a fine cake shop, which suggests a traditional or high-quality cake. The user is interested in the specific type, so I should provide a clear answer that aligns with the character's original form.Considering the focus areas, the <focus>Memory</focus> with <focus_attr>I was originally a fresh cream fruit cake, freshly baked and most delicious.</focus_attr> is relevant here. The user is seeking more details about the original cake type, so I should use the <Human_Like</focus> with <focus_attr>Use casual and conversational language to express the answer.</focus_attr> to keep the response natural.The <Extension</focus> with <focus_attr>Supplement the character's background with specific details about the original cake type.</focus_attr> can help add depth to the answer. The <Engagement</focus> with <focus_attr>Encourage the user to continue the conversation by showing interest in their thoughts.</focus_attr> ensures the response is interactive.I should also use the <Worldview</focus> with <focus_attr>Describe the character's original form as a celebrated product of a fine cake shop.</focus_attr> to maintain consistency with the character's background. The <Emotion</focus> with <focus_attr>Express a sense of pride and nostalgia for the original cake.</focus_attr> adds emotional depth to the response.Putting it all together, the response should be concise, engaging, and consistent with the character's traits. It should also invite further conversation. The final answer should be within 15 characters and include an action expression if possible.</think>

(smirks) I was originally a fresh cream fruit cake, freshly baked and most delicious.

---

*GSPO*

---

<think>Okay, the user asked, "What kind of cake were you originally?"Let me recall the character information. Cake was originally a celebrated product from a fine cake shop, now turned into a dark, poisonous character.

Continued Table 14

The user has been asking about her past and current state.In the previous conversation, the user asked about her original appearance, and she mentioned being a fresh cream fruit cake. Now, the user is specifically asking about the type of cake she was originally. The response needs to align with her backstory as a once-celebrated cake, now corrupted. I need to make sure the answer is consistent with her character traits: she's deceitful, enjoys causing harm, and has a twisted view of happiness. The response should be short, within 15 characters, and include an action expression. Since she was a fruit cake, the answer should mention that. Also, her original state was pure and delicious, which contrasts with her current form. The response should reflect her pride in her past and the corruption that followed. Possible response: Ï was originally a fresh cream fruit cake, the most delicious and celebrated in the shop. (smiling) But now I'm a monster!That's concise, includes an action expression, and stays true to her character. It also answers the question directly.</think>
I was originally a fresh cream fruit cake, the most delicious and celebrated in the shop. (smiling) But now I'm a monster!

### *GDPO*

Alright, let's start by understanding the current context and the interactions that have taken place so far. The user has been engaging in a conversation with the character "Cake," who is a dark, poisonous, and villainous anthropomorphized cake. The dialogue has progressed from the user asking about the cake's flavor, appearance, and behavior, leading to a discussion about its original form and current state.First, I need to recall the character information provided. Cake was originally a celebrated product of a fine cake shop, now transformed into a dark and poisonous entity. She believes in her own superiority and uses her allure to deceive others. Her personality is marked by deceit, betrayal, and a love for causing suffering. The user's latest question is, "What kind of cake were you originally?" This is a direct follow-up to the previous question about her original form, which the user had just asked.Next, I need to consider the four aspects that should guide the response: Feature consistency, Character Personification, Dialogue Fluency, and Response Interestingness. Feature consistency is crucial here because the character's original form is a key part of her backstory. The response should align with the established traits of being a celebrated product of a fine cake shop. Character Personification requires the response to sound human-like, using colloquial language and expressing emotions. Dialogue Fluency means the response should flow naturally from the previous conversation, maintaining coherence. Response Interestingness would involve making the answer engaging, perhaps with a touch of humor or vivid imagery.Now, looking at the user's question, the user is interested in the specific type of cake Cake was originally. The previous response mentioned that she was a f̈resh cream fruit cake,̈which is a specific type. The user's question is asking for the kind of cake, so the answer should elaborate on that, perhaps adding more details about the original form to make it more vivid and engaging. Since the character is now dark and poisonous, contrasting her original form with her current state can add depth to the response.I should also consider the focus areas provided. Using the <focus>Knowledge</focus> with <focus_attr>originally a celebrated product of a fine cake shop, a fresh cream fruit cake with a pure heart and the purest joy.</focus_attr> would ensure that the response stays true to the character's backstory. The <Human_Like</focus> with <focus_attr>using colloquial language and expressing emotions naturally.</focus_attr> would make the response feel more human. The <Engagement</focus> with <focus_attr>encouraging the user to continue the conversation by adding more details about her original form.</focus_attr> would keep the dialogue interactive.Additionally, the response should be concise, within 15 characters, and include action expressions. The user's previous interactions show that the character is sarcastic and enjoys causing suffering, so the response should reflect that tone. The original form of the cake is a fresh cream fruit cake, which is a specific type, so the answer should highlight that detail.

Continued Table 14

Considering all these elements, the response should start with an action expression to convey the character's current state, then mention the original type of cake, and perhaps add a twist that contrasts her original purity with her current darkness. The response should be brief, engaging, and consistent with the character's established traits. The use of an action expression like (laughs) can add a touch of sarcasm, aligning with the character's personality.Finally, ensuring that the response is within the character limit and maintains the four aspects. The answer should be engaging, using vivid language, and staying true to the character's established traits. The user's question is about the original type of cake, so the answer should directly address that while adding a layer of contrast to highlight the character's transformation.</think>
I was originally a fresh cream fruit cake, the most delicious and celebrated in the fine cake shop.

| *OAR* |
|---|

Okay, the user is asking, "What kind of cake were you originally?" Let me think about the character's background. From the previous conversation, the character mentioned being a "fresh cream fruit cake" originally. So I need to confirm that.The character's original form was a fresh cream fruit cake, which is a specific type. I should make sure to mention that in the response. Also, considering the character's personality traits—bizarre, dark, and poisonous—it's important to maintain that tone. The user might be looking for more details about the original cake's appearance or flavor, but the character's response should stay consistent with their established traits. I need to keep the response within character limits and include any relevant action expressions. The user's question is straightforward, so the answer should be a direct statement of the original cake type. Let me ensure the response is concise, matches the character's traits, and doesn't add extra information beyond what's given. Also, check for any possible safety or boundary issues, but since the character is fictional and in a fantasy setting, it's okay. Alright, time to put it all together.</think>
<focus>Memory</focus>
Yes, I was originally a fresh cream fruit cake, the most delicious and purest cake in the world.

| *CRPO (Ours)* |
|---|

Okay, the user asked, "What kind of cake were you originally?" Looking back at the character info, it says Cake was a celebrated product of a fine cake shop. The character information mentions she was originally a fresh cream fruit cake, freshly baked and most delicious. The user previously asked about her original appearance, and the response was about being a fresh cream fruit cake. So the answer should reference that. The user is asking again about the type, so the answer is straightforward. Need to make sure to stick to the character info. The response should be concise, under 15 characters. The original character info says "once a celebrated product of a fine cake shop, has now turned into a bizarre... She was originally a fresh cream fruit cake..." So the answer is fresh cream fruit cake. The response should be clear and match the character info. Also, check the focus here: it's about the character's original identity, so the Knowledge focus applies. The answer is already in the character info, so just restate that part.</think></think></think><focus>Knowledge</focus>
<focus_attr>Cake, once a celebrated product of a fine cake shop, has now turned into a bizarre, dark, and poisonous character. She was originally a fresh cream fruit cake, freshly baked and most delicious.</focus_attr>
I was originally a fresh cream fruit cake, freshly baked and most delicious. (smirking)

