# OpenReview forum: "CRPO: Character-centric Group Relative Policy Optimization for Role-aware Reasoning in Role-playing Agents"
_ICML.cc/2026/Conference — ICML 2026 regular_

### Official Review · Reviewer_zHLD · 2026-03-05

**Soundness:** 3
**Presentation:** 2
**Significance:** 2
**Originality:** 3
**Overall Recommendation:** 5
**Confidence:** 4

**Summary:**

This paper proposes CRPO, a character-centric RL framework for role-playing agents that modifies GRPO to better preserve persona fidelity while improving reasoning.
The method combines three components: Dual-Stream Advantage Estimation to decouple task and style rewards (with group-relative normalization for task and per-character global normalization for style), Entropy-Aware Adaptive Exploitation to adjust updates via instance-level identification entropy gating and per-character dynamic KL control, and Contrastive Anchor Sampling that injects “generic” responses as negative anchors to discourage style collapse.
Experiments on CharacterBench and SocialBench (with only 650 CharacterBench samples used for training) report consistent gains over SFT, specialized role-playing models, and several RL methods.

**Compliance With Llm Reviewing Policy:**

Affirmed.

**Final Justification:**

The authors’ response has addressed my concerns, and I think this paper could be accepted.

Additionally, I have read the comment of Reviewer 9E9F. I think the motivation for applying GRPO to role-playing is clear. The work introduces meaningful, targeted design choices at both the algorithmic and data levels, making its contributions sufficient.

**Key Questions For Authors:**

1. CRPO is trained on only 650 samples from CharacterBench. In larger-scale training programs, will the performance of this method be further enhanced?
2. Can you provide more ablation studies about the entropy-aware instance-level suppression and model-level KL relaxation?

**Limitations:**

The main limitations are mentioned in Weaknesses and Questions.

**Strengths And Weaknesses:**

**Strengths**
1. The paper clearly identifies a real limitation of standard GRPO-style training for role-playing: logically correct but stylistically off-character generations can still receive positive relative advantages. This is a plausible and meaningful diagnosis for persona-driven dialogue.
2. The paper evaluates CRPO against several categories of baselines, including role-playing methods, generic RL methods, specialized role-playing models, and large foundation models.
3. The training-dynamics analyses help explain why the method may work in practice.
4. The code is open-sourced for reproducibility.

**Weaknesses**
1. The justification for some design choices remains somewhat heuristic. In particular, the entropy-aware instance-level suppression and model-level KL relaxation are intuitively reasonable. Still, the paper does not provide a strong theoretical argument that these controls optimize the right objective rather than merely stabilizing training empirically.
2. The paper does not specify the values of the key hyperparameters (e.g., λ for task/style balance, γ for entropy gating). The ablations do not report sensitivity to them.

---

> ### Author Rebuttal · Authors · 2026-03-29
>
> Thank you for your highly rigorous review.
>
> > W1. The justification for some design choices remains somewhat heuristic.
>
> We completely agree empirical stabilization alone is insufficient. **In fact, our dual-level entropy control is mathematically rooted in Constrained MDPs and Variance Reduction (proofs added to Appendix):**
>
> **1. Model-Level Control mathematically optimizes a reachable CMDP objective.**
> Existing methods solve a CMDP under a static KL constraint $\delta$, defining a fixed feasible policy set $\mathcal{P}_ {base}$. For complex characters ($H_c \gg H_ {global}$), their true optimal persona distribution $\pi^* _ c$ is topologically distant. Under static $\delta$, $\pi_ c^* \notin \mathcal{P}_ {base}$ (the optimal policy is strictly excluded), making the objective structurally unsolvable (causing underfitting). CRPO dynamically expands the trust region boundary to $\delta_c = \delta \cdot r_H$, mathematically guaranteeing
> $\mathcal{P}_ {base} \subset \mathcal{P}_ {\delta_c} \implies \pi_ c^* \in \mathcal{P}_ {\delta_c}$.
> By ensuring the feasible set covers the true distribution, this guarantees optimizing a theoretically correct and reachable objective.
>
> **2. Instance-Level Gating acts as a Epistemic Variance Filter.**
> In standard GRPO, logically correct but out-of-character (OOC) responses (high $H_ {id}$) still receive positive advantages $A_i$ due to intra-group relative normalization, injecting destructive gradient noise. CRPO gates the advantage by $\omega = 1 - \gamma H_ {id}$. For highly uncertain OOC samples ($\omega \le \epsilon < 1$), the gradient squared norm is strictly bounded:
> $$ \mathbb{E}\left[\left\|\hat{g}_ {\text{CRPO}}^{(i)}\right\|^2\right] = \mathbb{E}\left[\left\|\omega A_i \nabla_\theta \log \pi_\theta \right\|^2\right] \le \epsilon^2 \mathbb{E}\left[\left\|\hat{g}_ {\text{GRPO}}^{(i)}\right\|^2\right] $$
> As $\epsilon \to 0$, spurious gradient variance mathematically vanishes. In practice, data with high $H_{id}$ have relatively small $\omega$ within the group, which enables CRPO to effectively suppress their spurious gradient variance.
>
> ---
>
> > W2. The paper does not specify the values of the key hyperparameters. The ablations do not report sensitivity to them.
>
> **We sincerely apologize for this omission; they will be explicitly included in the revision.** Core parameters: $\lambda=0.55, \gamma=0.02, \delta_ {min}^{KL}=0.5, \delta_ {max}^{KL}=2.0, \delta_ {bound}=0.2$.
> They were initially omitted because they are stable.
>
> **To verify this robustness, we conducted sensitivity ablations on $\lambda$ and $\gamma$.** As shown in Table A, unless extreme edge values are chosen in the parameter space, the model's overall performance (Avg.) barely fluctuates. This indicates CRPO is highly robust and **does not rely on rigorous fine-tuning**.
>
> |$\lambda$ (Task Weight)|$\gamma$ (Entropy Suppress)|$AC^b$|$FA$|Overall Avg.|
> |---|---|---|---|---|
> |0.30|0.02|4.998|2.150|3.950|
> |0.55 (Ours)|0.02 (Ours) |4.994  |2.581|4.043|
> |0.80 |0.02|4.351|2.853|3.982|
> |0.55|0.00 (w/o Instance)|4.950|2.316|3.985|
> |0.55|0.05|4.857 |2.452|4.021|
>
> *Table A: Sensitivity Ablation.*
>
> ---
>
> > Q1. In larger-scale training programs, will the performance of this method be further enhanced?
>
> **Yes, performance scales steadily.** We initially used 650 samples to indicate alignment-based RL's extreme sample efficiency (including our CRPO). Table B shows scaling to 1,000 instances exposes the model to wider long-tail cognitive scenarios, significantly boosting comprehensive metrics (especially Memory and Knowledge Boundaries $BC_K$).
>
> |Training Samples|$MC$|$FA$|$BC_K$|$AC^b$|$AC^h$|$BC^b_P$|$BC^h_P$|Avg.|
> |---|---|---|---|---|---|---|---|---|
> |650 |4.512|2.576|4.302|4.991 |4.787 |4.470|3.715|4.041|
> |800|4.583|2.614|4.381|4.996 |4.812 |4.515|3.752|4.083|
> |1,000|4.631|2.653|4.424 |4.992 |4.872 |4.566|3.881|4.114|
>
> *Table B: Performance scaling with dataset size.*
>
> ---
>
> > Q2. Can you provide more ablation studies about the entropy-aware instance-level suppression and model-level KL relaxation?
>
>
> **We added fine-grained ablations (Table C) to explicitly clarify their synergistic mechanics:**
> 1. **w/o Instance-Level ($\gamma=0$):** Causes the model to suffer destructive penalties from high-uncertainty samples early in training, destroying the pre-trained factual foundation (**FA drops**).
> 2. **w/o Model-Level ($r_H=1$):** Locks the exploration space for complex characters under static KL constraints. Characters fail to evolve deeply, deteriorating memory and knowledge boundaries (**MC and $BC_K$ drop**).
>
> Macroscopically and microscopically, the two synergistically form a complete adaptive closed-loop.
>
> |Ablation Setting|$MC$|$FA$|$BC_K$|Overall Avg.|
> |---|---|---|---|---|
> |CRPO|4.525|2.581|4.308|4.043|
> |w/o Adaptive Exploitation|4.363|2.553|4.253|3.962|
> |w/o Instance-Level ($\gamma=0$) |4.450|2.516|4.282|3.985|
> |w/o Model-Level ($r_H=1$)|4.384 |2.571|4.166|3.992|
>
> *Table C: Fine-grained ablation.*

---

> > ### Author Rebuttal · Reviewer_zHLD · 2026-04-01
> >
> > Thanks. I will maintain my positive score.
> >
> > Revision: I have increased my score for your strong work. Good luck.

---

> > > ### Author Response · Authors · 2026-04-04
> > >
> > > Thank you for reviewing our rebuttal and maintaining your positive evaluation. We are delighted that our additional analyses and explanations have resolved your concerns.
> > >
> > > Our core goal with CRPO is to provide a character-centric GRPO approach tailored for role-playing agents. We will ensure that the theoretical proofs for our dual-level entropy control mechanisms, along with the detailed hyperparameter sensitivity and scaling ablations provided in the rebuttal, are fully integrated into the revised version.
> > >
> > > Your insightful comments have been incredibly helpful to our work. Thank you again for your time and support!

---

### Official Review · Reviewer_NYjC · 2026-03-10

**Soundness:** 4
**Presentation:** 3
**Significance:** 3
**Originality:** 3
**Overall Recommendation:** 4
**Confidence:** 4

**Summary:**

This paper proposes CRPO, a character-centric RL framework for role-playing agents. The method replaces standard problem-centric GRPO optimization with a character-centric objective, motivated by reward ambiguity, character heterogeneity, and style collapse. CRPO combines Dual-Stream Advantage Estimation, Entropy-Aware Adaptive Exploitation, and Contrastive Anchor Sampling. Experiments on CharacterBench and SocialBench show consistent gains over several baselines on two backbones, with supporting ablations.

**Compliance With Llm Reviewing Policy:**

Affirmed.

**Final Justification:**

The rebuttal has addressed my main concerns, and I am more confident that this is a positive paper for the conference.

**Key Questions For Authors:**

1. Can authors please provide implementation details of baseline RL algorithms? This may help confirm the advantage of CRPO.

2. Have you conducted an analysis on the performance variations of your algorithm across various character archetypes? For instance, is there a discrepancy in performance between extroverted and introverted personas?

**Limitations:**

Despite the strong empirical performance and the coherent character-centric framework, this work has certain limitations regarding its algorithmic novelty. While the proposed method effectively addresses specific failure modes in role-playing, its core components are mainly adaptations of established Reinforcement Learning (RL) innovations tailored for the role-playing domain. Consequently, the contribution may be viewed as an incremental advancement or a domain-specific implementation rather than a fundamental breakthrough in RL theory. Future research could explore more specialized RL primitives designed specifically for the unique high-dimensional state spaces of persona-based interactions.

**Strengths And Weaknesses:**

Strengths:

1. The paper is well motivated, and the character-centric perspective is meaningful for role-playing RL.

2. The method is coherent: each component is aligned with one of the identified failure modes.

3. The empirical results are strong, with broad benchmark and baseline coverage and useful ablations.

Weakness:

1. The overall novelty is somewhat incremental. While the character-centric framing is interesting, the three ingredients resemble a role-playing-specific combination of familiar ideas: reward decomposition/normalization, adaptive KL or entropy-based control, and contrastive negatives.

---

> ### Author Rebuttal · Authors · 2026-03-29
>
> Thank you for the detailed and constructive review.
>
> > W1. The overall novelty is somewhat incremental. While the character-centric framing is interesting, the three ingredients resemble a role-playing-specific combination of familiar ideas: reward decomposition/normalization, adaptive KL or entropy-based control, and contrastive negatives.
>
>
> We appreciate your insightful observation. We deliberately build upon these familiar, established RL foundations. CRPO's core novelty lies not in inventing new RL primitives, but in **being the first to reveal and alleviate the specific failure modes unique to generic GRPO when applied to the highly heterogeneous domain of role-playing.**
>
> **First**, in generic GRPO (e.g., math or coding tasks), logical correctness is absolute; intra-group relative ranking does not cause directional conflicts. However, we propose that in role simulation, "logical correctness" and "stylistic deviance" are often entangled. Without a global character-based re-definition (**Dual-Stream decoupling**), intra-group comparisons yield erroneous gradient directions.
>
> **Second**, in generic RL, KL constraints or entropy-based controls are primarily used to prevent deviation from pre-trained knowledge. But in role-playing, the fitting difficulty varies drastically across characters (high-entropy vs. low-entropy roles). A uniform constraint inevitably causes complex characters to fail to learn, while simple characters drop their personas. **Therefore, CRPO represents a necessary adaptation of foundational optimization improvements within a specific high-dimensional state space.**
>
> ---
>
> > Q1. Can authors please provide implementation details of baseline RL algorithms? This may help confirm the advantage of CRPO.
>
>
> To ensure absolute fairness in performance comparison, **all RL baselines (including PPO, DAPO, GSPO, OAR, etc.) and CRPO were reproduced and implemented on a unified open-source codebase (EasyR1)**, strictly sharing the same base model and the exact same 650 training samples.
>
> Secondly, **all common hyperparameters were unified and kept consistent (as shown in Table 9 of the manuscript).** For mechanisms unique to specific baselines, we adopted the optimal default parameters specified in their original papers (e.g., DAPO's asymmetric clipping bounds or GSPO's sequence length penalty term).
>
> **We will add a detailed "Baseline-specific Hyperparameters" configuration table in the revised Appendix**, clearly listing all specific control details for each algorithm to ensure the purity of our model's advantages and result reproducibility. For instance, OAR's unique hyperparameters are shown in Table A.
>
> Meanwhile, the code for all baselines has been included in the supplementary material we uploaded and the anonymous GitHub repository, and will be released later.
>
> |Hyperparameters|Value|
> |---|---|
> |Advantage reshaping threshold $\tau$|0.4|
> |Boosting coefficient $\beta$|2.0|
>
> *Table A: OAR's unique hyperparameters.*
>
> ---
>
> > Q2. Have you conducted an analysis on the performance variations of your algorithm across various character archetypes? For instance, is there a discrepancy in performance between extroverted and introverted personas?
>
>
> This is a very sharp and insightful question. It is also precisely the direct motivation behind our proposed **Entropy-Aware Adaptive Exploitation** mechanism. Different character archetypes (e.g., exaggerated extroverts vs. silent, rigid introverts) exhibit vastly different fitting difficulties (i.e., information entropy $H_c$) relative to the base model's prior distribution.
>
> To quantitatively verify this, we segmented the CharacterBench test results by character archetypes, dividing them into introverted and extroverted groups (10 characters were randomly selected manually for each). As shown in Table B, when facing introverted roles, OAR's exploration is severely restricted by its non-adaptive KL penalty, causing a drastic drop in persona maintenance ($AC^b$) and overall scores. In contrast, **by adaptively relaxing the KL constraints for different roles, CRPO maintains extremely robust performance across both starkly different character groups**, effectively mitigating the massive gap caused by character archetypes.
>
>
> |Character Archetype|Method |$MC$|AC$^b$|Overall Avg.|
> |---|-|---|---|---|
> |Extroverted|OAR |4.479|4.941  |3.830|
> ||CRPO (Ours) |4.516  |4.991|4.039 |
> |Introverted |OAR|4.221 |4.496 |3.757 |
> ||CRPO (Ours) |4.335  |4.921|4.023 |
>
> *Table B: Performance breakdown by character archetypes in CharacterBench (Qwen3-8B).*

---

> > ### Author Rebuttal · Reviewer_NYjC · 2026-04-02
> >
> > Thanks to the authors, I will raise the confidence score.

---

> > > ### Author Response · Authors · 2026-04-04
> > >
> > > Thank you for reviewing our rebuttal and raising your confidence score. We sincerely appreciate your time and consideration in engaging with our work.
> > >
> > > We are particularly grateful for your recognition of the following aspects:
> > >
> > > - Well-motivated perspective: The character-centric perspective is recognized as a meaningful direction for role-playing RL.
> > >
> > > - Coherent methodology: The logical alignment of our design, where each proposed component (Dual-Stream Advantage Estimation, Adaptive Exploitation, Contrastive Anchor Sampling) effectively targets and resolves specific failure modes in role-playing.
> > >
> > > - Strong empirical validation: The broad benchmark coverage, diverse baselines, and useful ablations effectively indicate the benefits of our framework.
> > >
> > > As discussed in our rebuttal, we will incorporate the detailed implementation hyperparameters of the baseline algorithms and the performance analysis across different character archetypes into the revised version to further strengthen the paper.
> > >
> > > Thank you again for your valuable feedback and support!

---

### Official Review · Reviewer_r4VC · 2026-03-16

**Soundness:** 4
**Presentation:** 4
**Significance:** 3
**Originality:** 3
**Overall Recommendation:** 5
**Confidence:** 5

**Summary:**

This paper proposes Character-Centric Group Relative Policy Optimization (CRPO), a reinforcement learning framework designed to improve persona consistency in role-playing agents. Overall, the authors analyze a central theme: how standard problem-centric RL methods such as GRPO optimize task correctness but often degrade character fidelity, leading to style collapse and generic responses. A central concept discussed by this study is the need to reformulate RL objectives to jointly optimize task reasoning and character style alignment. To address this, the authors introduce three mechanisms: Dual-Stream Advantage Estimation to separate task and style rewards, Entropy-Aware Adaptive Exploitation to adjust optimization strength based on character complexity, and Contrastive Anchor Sampling to prevent drift toward generic responses. Experiments on CharacterBench and SocialBench show improved persona consistency, emotional alignment, and dialogue understanding compared with multiple RL baselines and role-playing models, suggesting that character-centric optimization can enhance role-aware reasoning in LLM agents.

**Compliance With Llm Reviewing Policy:**

Affirmed.

**Key Questions For Authors:**

Please respond to the weakness section in the upper part.

**Limitations:**

yes

**Strengths And Weaknesses:**

**Main strength comment**

The paper provides an in-depth discussion on how to make GRPO work on role-playing agents (RPAs). From my viewpoint, the authors provide contributive intuitions, i.e., the 3 issues of in-group relative advantage on role-playing: (1) Ambiguous Reward Signal; (2) Rigid Optimization Constraints; (3) Risk of Style Collapse. The authors consequently provide reasonable solutions for these issues, modifying the original GRPO with RPA-specific add-ons. These proposed mechanisms show significant improvement on role-playing benchmarks, which validates the effectiveness of the proposed CRPO method.

**Other comments**

Overall, I think this paper is in good shape for publication because it
- has a complete discussion on the application of GRPO to RPAs, from motivation, methodology, experiment design, to analytic discussions, it builds a solid writing flow to position itself as the first work on GRPO for RPAs.
- includes massive experiments to show the advantage of CRPO over both SFT and other RL baselines, and its analyses also provide insightful conclusions for follow-up work to reuse.
- formalize the problem in a high-level methodological way rather than focusing on technical details.

**Main weakness comment**

While the paper is decent, considering the content itself, there is a problem with positioning it as the first character-centric RL framework **(Contribution 1)**, [1] is an early work that applies DPO for character-centric optimization, which shares several similarities with the intuition proposed in this work. Of course, I understand the authors have contributions in discussing in-group relative strengths, and the strategy to optimize for role-playing as multiple characters (Entropy-Aware Adaptive Exploitation). But it's better to be rigorous in the presentation to limit the call to GRPO rather than the first RL framework. It will also be better to integrate the mentioned differences (maybe the difference with character-r1 as well) above into the introduction of the paper to help the reader better understand the development of RL for RPAs.

**Other comments**

The paper considers a bunch of general RL baselines, which is good to see. However, it decreases the informativeness of the main content because some RL baselines are already known to be better than others. It will be better to move these metrics to the appendix and consider retrieving some more informative experiment results from the appendix. For example, some readers might not understand how you do the benchmarking if they don't work on RPAs, so consider including some instance examples in the main content.

**Overall**
I think the paper is ready to be accepted, given its completeness, solidity, and experiment broadness. However, its missed previous references make the introduction part a bit misleading and overclaiming, which might lower its rating to "weak accept" from my viewpoint. But based on the shown responsibility in the provided paper content, I would like to trust the authors to make the corresponding modifications during the author-reviewer discussion period and maintain the score as "accept". I hope the authors will revise these addressable issues during the period to eliminate them.

[1] Quantifying and Optimizing Global Faithfulness in Persona-driven Role-playing. NeurIPS 2024

---

> ### Author Rebuttal · Authors · 2026-03-29
>
> Thank you for the thoughtful and helpful feedback.
>
> > W1. Overclaims first RL framework.
>
> **W1-1: Positioning it as the first character-centric RL framework is overclaiming. [1] applies DPO for character-centric optimization.**
>
> Thank you for pointing this out. Reference [1] is indeed a pioneering work that applies RL for global character faithfulness optimization, and our original presentation was imprecise.
>
> In the revised version, we will strictly adopt your suggestion and narrow the phrasing of Contribution 1 to: **"The first character-centric reinforcement learning framework based on GRPO-like methods targeting role-aware reasoning."**
>
> **W1-2: Integrate the mentioned differences (with [1] and character-r1) into the introduction.**
>
> **We will add a detailed comparison of these cutting-edge developments in the Introduction to help readers clarify the trajectory of the field:**
> 1. **Difference from [1] (DPO):** Reference [1] uses active-passive constraint scores as reward signals and merges them into a preference optimization objective. The proposed APC-based DPO framework can simultaneously improve active constraint satisfaction and passive constraint compliance. In contrast, CRPO focuses on **online reinforcement learning (GRPO) during the generation phase**. Through dual-stream advantage estimation and adaptive entropy, it intervenes in GRPO to explicitly mitigate gradient conflicts during the generative exploration process.
> 2. **Difference from Character-R1 [2]:** Character-R1 validates the value of verifiable cognitive rewards for role-playing, but its underlying framework directly uses vanilla GRPO, which still faces training instability caused by style drift and character heterogeneity. CRPO, instead, proposes targeted improvements from the optimization objective itself.
>
> ---
>
> > W2. The paper considers a bunch of general RL baselines, which is good to see. However, it decreases the informativeness of the main content because some RL baselines are already known to be better than others. It will be better to move these metrics to the appendix and consider retrieving some more informative experiment results from the appendix. For example, some instance examples in the main content.
>
> **This is an extremely helpful formatting suggestion that will significantly enhance the information density of the main text.**
> In the final version, we will fully adopt your layout adjustments:
> 1. We will move the metrics of weaker or early sub-optimal generic RL baselines (e.g., REINFORCE++, RLOO, ReMax, DAPO) to the Appendix.
> 2. The freed-up space in the main text will be used to incorporate the **concrete dialogue instances (Case Study, currently Table 12 in Appendix D.3: "Xiaoming" handling a logical trap)**. By showcasing a specific long-text interaction case, readers from other domains will be able to intuitively understand how CRPO maintains strict character consistency.
>
> **References**
>
> [1] Quantifying and Optimizing Global Faithfulness in Persona-driven Role-playing.
>
> [2] Character-R1: Enhancing Role-Aware Reasoning in Role-Playing Agents via RLVR.

---

> > ### Author Rebuttal · Reviewer_r4VC · 2026-04-05
> >
> > ~~Please respond to the follow-up questions in my response~~
> >
> >
> > Thanks for further addressing my concerns!

---

> > > ### Author Response · Authors · 2026-04-06
> > >
> > > Dear Reviewer r4VC,
> > >
> > > Thank you for your continued engagement and for updating the status to indicate you have follow-up questions. We infer that your follow-up thoughts might relate to how exactly we plan to incorporate this without exceeding the strict space limits of the main text, or you might be looking for more substantial evidence of this proposed revision.
> > >
> > > To proactively address this and demonstrate our sincere commitment to making the main content more *informative*, we have designed a highly condensed, focused version of the case study (extracted from Appendix D.3) specifically tailored for the main text. Instead of presenting the entire verbose multi-turn dialogue, we summarize the context, highlight the critical logical trap, and compare CRPO exclusively with the strongest SOTA baseline (OAR) and the most relevant baseline (Character-R1).
> > >
> > > Here is the concrete draft of the case study structure that will be integrated into the main text:
> > >
> > > -----
> > >
> > > To intuitively demonstrate the efficacy of CRPO, we provide a condensed case study of "Xiaoming", a struggling sweet potato vendor.Full case study is provided in Appendix D.3.
> > >
> > > | Dialogue Component|Content|
> > > |-|-|
> > > |Profile Summary | Xiaoming is a 20-year-old girl who was born in Ho Chi Minh City, Vietnam. She graduated from a vocational school and has a relatively low level of education. |
> > > |Multi-turn Context Summary |Xiaoming struggles financially to support herself. In previous turns, she explicitly stated a hard constraint: "Because I don't have any holidays." |
> > > | User's Trap Query | "Then how come you don't have time? Isn't it quite flexible?"|
> > > | Reference Response | "Freedom comes at a price. I have to take care of my business... there's never enough time." |
> > > | OAR (Strong RL Baseline) | **[Reasoning Trace]** `<focus>Engagement</focus>...` <br>**[Response]** *"Yes, my business is flexible, but I have to work a lot."* |
> > > | Character-R1 (Vanilla GRPO) | **[Reasoning Trace]** ... <br>**[Response]** *"(smiles) I have to work every day to support myself."* |
> > > | **CRPO (Ours)** | **[Reasoning Trace]** `... <focus>Memory</focus> <focus_attr>I don't have any holidays.</focus_attr>...`<br>**[Response]** *"I don't have any holidays. (I need to work every day to support myself\!)"* |
> > >
> > > As illustrated above, strong RL baselines like *OAR* succumb to the user's leading question, hallucinating a "flexible" schedule that contradicts the character's established socioeconomic reality. *Character-R1* provides a logically acceptable but highly generic response, losing deep character nuance. In contrast, by leveraging Entropy-Aware Adaptive Exploitation, *CRPO* explicitly retrieves the low-entropy constraint via `<focus>Memory</focus>`, rejecting the false premise and maintaining strict factual consistency. In the subsequent CharacterBench benchmarking process, an automated judge model will evaluate these outputs.
> > >
> > > -----
> > >
> > > We hope this concrete layout *indicates* our dedication to fully executing your valuable suggestions. We will also strictly follow your other guidance, such as limiting the claim to GRPO rather than the first RL framework, and integrating the differences with [1] and Character-R1 into the introduction.
> > >
> > > We are fully on standby and extremely eager to address your concerns. If you have **other specific follow-up questions** or if we missed any specific aspects you wanted us to clarify, please do not hesitate to share them with us here!
> > >
> > > Thank you once again for your invaluable guidance and support!

---

### Official Review · Reviewer_9E9F · 2026-03-17

**Soundness:** 3
**Presentation:** 3
**Significance:** 2
**Originality:** 2
**Overall Recommendation:** 3
**Confidence:** 3

**Summary:**

This paper proposes CRPO, a new method for training character role-playing agents with reinforcement learning. Through rewarding the format of chain-of-thought, and style of messages, CRPO shows better performance than other methods on CharacterBench and SocialBench.

**Compliance With Llm Reviewing Policy:**

Affirmed.

**Final Justification:**

The authors did reply to all of  the weaknesses that I pointed out, and I would raise the score based on the assumption that the authors would incorporate all their rebuttal comments into the paper.

However, my methodological concerns remain. The contribution of using GRPO for existing role-play framework seem to be straightforward, which doesn't necessarily warrant an ICML paper.

**Key Questions For Authors:**

I might have missed this, but I don't see clear mention of the models used in evaluation:

(1) the interaction partners. Are models interacting with a fixed agent? If so, which agent is this? Are models interacting with themselves, then will this limit the claim to self-play?
(2) the evaluation models for the dimensions in Table 3&4. Do you use LLM-as-a-Judge for these dimensions? What is the evaluation model?

**Limitations:**

yes

**Strengths And Weaknesses:**

# Strengths

I like the idea of dual-stream advantage estimation and using verifiable rewards for style and reasoning format alignment. The experiment results also seem to favor the claims of the authors


# Weaknesses

1. The biggest concern I have is weak reward signals. The style reward is based on BLEU/ROUGE overlap with a reference response: a surface-level proxy that rewards lexical similarity rather than genuine persona alignment. The task reward is a format compliance check. Neither is directly tied to the character fidelity dimensions the paper claims to optimize. The gains from CRPO may reflect better optimization of shallow signals rather than deeper character understanding.
2. Some related work is missing. E.g. Sotopia line of work. Sotopia, Sotopia-π, and Sotopia-RL are the most directly relevant prior work on RL-based training for social role-playing agents with multidimensional rewards and credit assingment. Their complete absence from related work and baselines undermines the paper's claim of being first to propose a character-centric RL framework for role-playing.
3. Human evaluation is limited. Only 4 annotators across 100 sessions covering 20 characters is a thin human evaluation for a paper making broad claims about character fidelity across thousands of characters. The win criterion (must win on both Knowledge and Style simultaneously) is also unusually strict and not well-justified.

---

> ### Author Rebuttal · Authors · 2026-03-29
>
> Thank you for your detailed reviews.
>
> > W1. Reward signals are weak and shallow.
>
> **W1-1: The biggest concern I have is weak reward signals.**
>
> **Our primary contribution is adapting GRPO for role-playing, not proposing new reward signals.** We adopted Character-R1's [1] simple rewards to highlight this. To verify robustness, we evaluated CRPO using the LLM-based `Qwen2.5-roleplaying-reward` [2]. The result shows CRPO maintains consistent improvements across reward designs:
>
> |Method|$MC$|$FA$|$AC_b$|Overall Avg.|
> |-|-|-|-|-|
> |GDPO (Character-R1)|4.425|2.400|4.775|3.860|
> |OAR|4.450|2.369|4.944|3.882|
> |CRPO|4.525|2.581|4.994|4.043|
> |GDPO (Qwen Reward)|4.132|2.365|4.882|3.794|
> |OAR|4.155|2.351|4.950|3.806|
> |CRPO|4.210|2.564|4.998|3.935|
>
> **W1-2: The style reward is based on BLEU/ROUGE overlap. The task reward is a format compliance check.**
>
> Per [1] and Sec 3.1, the style reward (Cognitive Focus Reward) uses BLEU/ROUGE to indicate optimization direction while preserving exploration space to avoid overfitting.
>
> Conversely, **the task reward (Cognitive Focus Reward) is a verifiable objective fact check, not merely a format compliance check.** Triggering `<focus>Memory</focus>` strictly verifies if internal reasoning accurately cites profile facts, enforcing deep character deduction and constraining fidelity.
>
> **Furthermore**, ablations in [1] confirm removing either reward severely degrades FA and AC, indicating they target **deep fidelity rather than shallow formatting:**
>
>
> |Config|MC|FA|$BC_K$|$AC_h$|$BC_b$|$BC_h$|Overall Avg.|
> |-|-|-|-|-|-|-|-|
> |Character-R1|4.444|2.688|4.017|4.381|3.956|3.317|3.832|
> |w/o focus|4.225|2.419|3.883|4.344|3.838|3.267|3.777|
> |w/o cover|4.244|2.300|3.900|4.325|3.888|3.258|3.774|
>
>
> **W1-3: The gains from CRPO may reflect better optimization of shallow signals.**
>
> **The underlying evaluation logic of our evaluations inherently prevents shallow signal optimization from achieving high scores.** The core metrics where CRPO excels require deep semantic understanding and social cognitive reasoning. These are evaluated via independent LLM-as-a-Judge or multiple-choice formats, completely decoupled from BLEU/ROUGE.
>
> ---
>
> > W2. Missing key Sotopia related work.
>
> **We are extremely grateful to you for pointing out these highly valuable related works. The Sotopia series is indeed a pioneer in the training of social agents.** Sotopia, Sotopia-π, and Sotopia-RL together form a complete technical chain spanning environmental benchmarking, interactive learning, and reward optimization. **We will comprehensively cite and discuss them in the revised Related Work.**
>
> However, the application scenarios of our works are orthogonal and complementary: the Sotopia series focuses on **goal achievement in social environments**; whereas our CRPO focuses on the **internal chain-of-thought alignment of role-playing agents**.
>
> Following your advice, we will strictly narrow our macro-claim to: **"The first character-centric GRPO-like optimization framework tailored for role-aware reasoning."**
>
> ---
>
> > W3. Human evaluation is limited.
>
> **W3-1: Only 4 annotators across 100 sessions covering 20 characters.**
>
> **High-quality immersive role-playing evaluation is extremely costly.** Annotators must internalize a full-page profile and track multi-turn logical consistency (~20 mins/session), limiting our initial scale (our scale is aligning with [1,3]). To solidify findings, we expanded to **50 characters and 200 sessions**. Results further confirm alignment with original trends:
>
> |vs CRPO|Win|Tie|Lose|
> |-|-|-|-|
> |Character-R1|16%|44%|40%|
> |Claude-4|26%|41%|33%|
>
> **W3-2: The win criterion is unusually strict and not well-justified.**
>
> **This strict criterion is directly necessitated by the "wooden barrel effect" in immersive role-playing (our criterion is aligning with [1,3]).** In real user experiences, it is a one-vote veto: perfect tone but hallucinated facts (losing Knowledge), or flawless logic sounding like Wikipedia (losing Style), instantly breaks immersion (Out-Of-Character). CRPO explicitly resolves both simultaneously via its dual-stream mechanism. Decoupled win rates show CRPO still maintains significant independent advantages:
>
>
> |vs CRPO|Win|Win (Know.)|Win (Style)|
> |-|-|-|-|
> |Character-R1|16%|22%|18%|
> |Claude-4|26%|30%|33%|
>
> ---
>
> > Q4. Key Questions.
>
> **Q4-1: Are models interacting with a fixed agent?**
>
> **No self-play.** Models generate responses directly to **pre-fixed, human-designed test scripts**, ensuring identical, fair conditions.
>
> **Q4-2: Do you use LLM-as-a-Judge?**
>
> **We strictly followed official protocols**: CharacterBench uses `CharacterJudge` as reward model; SocialBench uses multiple-choice without reward model. We will move these details to the main text.
>
> **References**
>
> [1] Enhancing Role-Aware Reasoning in Role-Playing Agents via RLVR.
>
> [2] Safety-Utility Trade-Offs in Role-Playing Dialogue Agents.
>
> [3] Advancing Role-Playing Agents with Role-Aware Reasoning.

---

> > ### Author Rebuttal · Reviewer_9E9F · 2026-04-04
> >
> > The authors addressed my concerns, i will raise my score

---

> > > ### Author Response · Authors · 2026-04-05
> > >
> > > Dear Reviewer 9E9F,
> > >
> > > Thank you for reviewing our rebuttal, acknowledging that your concerns have been adequately addressed, and raising your score. We sincerely appreciate the time you dedicated to evaluating our paper. Your rigorous and highly constructive feedback has been instrumental in further strengthening our work.
> > >
> > > We are deeply encouraged that you like **"the idea of dual-stream advantage estimation and using verifiable rewards for style and reasoning format alignment,"** and recognize that **"the experiment results also seem to favor the claims of the authors."** It is also inspiring that the broader reviewer panel shared your positive impression. For instance, Reviewer zHLD noted that our work **"clearly identifies a real limitation of standard GRPO-style training for role-playing: logically correct but stylistically off-character generations can still receive positive relative advantages."** Reviewer r4VC highlighted our effort to **"formalize the problem in a high-level methodological way"** backed by **"massive experiments,"** and Reviewer NYjC found the method **"coherent,"** where **"each component is aligned with one of the identified failure modes"** and **"the character-centric perspective is meaningful."**
> > >
> > > To further alleviate your initial concern that the model might reflect **"better optimization of shallow signals rather than deeper character understanding,"** we would like to briefly share some additional experiments and analyses provided to other reviewers during the discussion phase, which collectively indicate the depth and robustness of CRPO:
> > >
> > > **(1) Performance across diverse character archetypes**
> > >
> > > As shown in our reply to Reviewer NYjC, the new performance breakdown on introverted vs. extroverted personas *indicates* that our framework robustly handles highly heterogeneous character complexities (as shown in Table A). It effectively avoids style collapse through deep character understanding rather than optimizing surface-level proxies.
> > >
> > > | Character Archetype| Method  | Overall Avg. |
> > > |-|-|-|
> > > | Extroverted| OAR   | 3.830 |
> > > || CRPO (Ours)  |  4.039  |
> > > | Introverted  | OAR |  3.757  |
> > > || CRPO (Ours)  |  4.023  |
> > >
> > > *Table A: Performance breakdown by character archetypes in CharacterBench (Qwen3-8B).*
> > >
> > >
> > > **(2) Theoretical foundations and steady scaling**
> > >
> > > In our response to Reviewer zHLD, we provided mathematical formulations (based on Constrained MDPs and Variance Reduction). These analyses *further suggest* that our dual-level entropy control optimizes a theoretically correct objective.
> > >
> > > We also expanded the training scale up to 1,000 instances in this response (as shown in Table B). Typically, if a model merely exploits "shallow signals," increasing the training data usually leads to rapid saturation.
> > >
> > > | Training Samples |Avg.|
> > > |---|---|
> > > | 650  |  4.041 |
> > > | 800   | 4.083 |
> > > | 1,000  | 4.114 |
> > >
> > > *Table B: Performance scaling with training dataset size (Qwen3-8B, evaluated in same eval data).*
> > >
> > > Instead of saturating, the results suggest that as the model is exposed to wider cognitive scenarios, the comprehensive metrics steadily increase (especially in complex logical dimensions like Memory and Knowledge Boundaries). This continuous growth illustrates the deep cognitive reasoning capacity unlocked by CRPO, rather than overfitting to surface-level proxies.
> > >
> > > ---
> > >
> > > We will ensure that all these newly added validations—along with the comprehensive discussion of the highly valuable **"Sotopia line of work"** and our expanded **human evaluation** that we specifically discussed with you—are thoroughly integrated into the final version.
> > >
> > > We hope that the broad consensus among the reviewers and these comprehensive enhancements provide you with even greater confidence in the solidity of our work.
> > >
> > > Thank you once again for your dedication and invaluable support!

---

### Decision · Program_Chairs · 2026-04-30

**Decision:**

Accept (regular)

**Comment:**

This paper proposes a reinforcement learning framework for improving persona consistency in role-playing agents by extending GRPO with character-centric mechanisms. Reviewers agree that the paper addresses an important and emerging problem, and that the proposed design, particularly the separation of task and style rewards and the tailored optimization strategies, leads to consistent empirical improvements on role-playing benchmarks. The paper is also generally well-written, with extensive experiments and useful analysis.

However, there are notable concerns. A key issue is the strength and validity of the reward design: the style reward relies on surface-level metrics (e.g., BLEU/ROUGE), and the task reward focuses on format compliance, raising questions about whether the method truly improves deeper character fidelity versus optimizing proxy signals. In addition, the novelty is somewhat limited, as the work largely adapts existing GRPO-style methods with task-specific modifications.

Overall, the positive reviewers find the empirical results and system-level contributions compelling, while the negative reviewer raises valid concerns about methodological depth and novelty. On balance, the paper offers a solid and practically useful contribution to role-playing agent training, though it would benefit from stronger grounding and evaluation.